# Robustness and parameter geography in post-translational modification systems

**Kee-Myoung Nam**[1], **Benjamin M. Gyori**[2], **Silviana V. Amethyst**[3], **Daniel J. Bates**[4], **Jeremy Gunawardena**[1] *

**1** Department of Systems Biology, Harvard Medical School, Boston, Massachusetts, United States of America, **2** Laboratory of Systems Pharmacology, Harvard Medical School, Boston, Massachusetts, United States of America, **3** Department of Mathematics, University of Wisconsin–Eau Claire, Eau Claire, Wisconsin, United States of America, **4** Department of Mathematics, United States Naval Academy, Annapolis, Maryland, United States of America

\* jeremy@hms.harvard.edu

**Data Availability Statement:** All datasets referenced in the paper (outside of the files in S1 Dataset) are available on Mendeley Data. Their DOIs are listed in S1 File.

## Abstract

Biological systems are acknowledged to be robust to perturbations but a rigorous understanding of this has been elusive. In a mathematical model, perturbations often exert their effect through parameters, so sizes and shapes of parametric regions offer an integrated global estimate of robustness. Here, we explore this "parameter geography" for bistability in post-translational modification (PTM) systems. We use the previously developed "linear framework" for timescale separation to describe the steady-states of a two-site PTM system as the solutions of two polynomial equations in two variables, with eight non-dimensional parameters. Importantly, this approach allows us to accommodate enzyme mechanisms of arbitrary complexity beyond the conventional Michaelis-Menten scheme, which unrealistically forbids product rebinding. We further use the numerical algebraic geometry tools Bertini, Paramotopy, and alphaCertified to statistically assess the solutions to these equations at $\sim 10^9$ parameter points in total. Subject to sampling limitations, we find no bistability when substrate amount is below a threshold relative to enzyme amounts. As substrate increases, the bistable region acquires 8-dimensional volume which increases in an apparently monotonic and sigmoidal manner towards saturation. The region remains connected but not convex, albeit with a high visibility ratio. Surprisingly, the saturating bistable region occupies a much smaller proportion of the sampling domain under mechanistic assumptions more realistic than the Michaelis-Menten scheme. We find that bistability is compromised by product rebinding and that unrealistic assumptions on enzyme mechanisms have obscured its parametric rarity. The apparent monotonic increase in volume of the bistable region remains perplexing because the region itself does not grow monotonically: parameter points can move back and forth between monostability and bistability. We suggest mathematical conjectures and questions arising from these findings. Advances in theory and software now permit insights into parameter geography to be uncovered by high-dimensional, data-centric analysis.

**Funding:** K-MN and JG were supported by National Science Foundation award #1462629 (https://www.nsf.gov/). DJB was supported by National Science Foundation award #1719658. SVA and DJB were supported by National Science Foundation award #1115668. The funders had no role in study design, data collection and analysis, decision to publish, or preparation of the manuscript.

**Competing interests:** The authors have declared that no competing interests exist.

## Author summary

Biological organisms are often said to have robust properties but it is difficult to understand how such robustness arises from molecular interactions. Here, we use a mathematical model to study how the molecular mechanism of protein modification exhibits the property of multiple internal states, which has been suggested to underlie memory and decision making. The robustness of this property is revealed by the size and shape, or "geography," of the parametric region in which the property holds. We use advances in reducing model complexity and in rapidly solving the underlying equations, to extensively sample parameter points in an 8-dimensional space. We find that under realistic molecular assumptions the size of the region is surprisingly small, suggesting that generating multiple internal states with such a mechanism is much harder than expected. While the shape of the region appears straightforward, we find surprising complexity in how the region grows with increasing amounts of the modified substrate. Our approach uses statistical analysis of data generated from a model, rather than from experiments, but leads to precise mathematical conjectures about parameter geography and biological robustness.

## Introduction

Biological systems are widely acknowledged to be robust, which informally means that some property of a system is insensitive to perturbations. Particular forms of robustness, such as homeostasis in physiology [1], canalisation in development [2], and resilience in ecology [3], have been extensively studied. Robust design has been suggested as a general biological criterion [4] with parallels to engineering [5, 6] and as an important requirement for synthetic biological systems [7]. Furthermore, organismal robustness is often invoked as a form of buffering, to account for the extensive genotypic variation seen in populations, on which natural selection may subsequently act [8, 9]. A better understanding of robustness is therefore relevant to many aspects of biology.

We approach this problem here through mathematical analysis. The perturbations to which a biological system is robust typically arise in the system's environment. When the system is represented by a mathematical model, it is the model's parameters which capture the interactions between the system and its environment, so that perturbations are represented by changes in parameter values. This kind of parametric robustness is not the only way in which robustness can be interpreted mathematically—the effect of noise on the dynamics or of changes to conserved quantities may also be important [10]—but parametric robustness has been widely studied.

To clarify this kind of parametric robustness further, it is helpful to keep in mind the relationship between parameters and state variables, as shown in Fig 1A. We have assumed that the underlying mathematical model is that of a system of ordinary differential equations, because we will use this kind of model here, but a similar picture could be drawn for a system of difference equations, or for stochastic or partial differential equations. It is only when the parameters are given numerical values that a dynamics is specified in the state space. If the state variables are in some initial condition, the system follows a trajectory over time and eventually reaches a steady-state or a limit cycle or some more complicated attractor [11]. Crucially, this dynamics in the state space depends on the choice of numerical values in the parameter space. We typically expect the parameter space to break up into regions so that the dynamical portrait varies only <u>quantitatively within</u> a region, but changes <u>qualitatively between</u> regions.

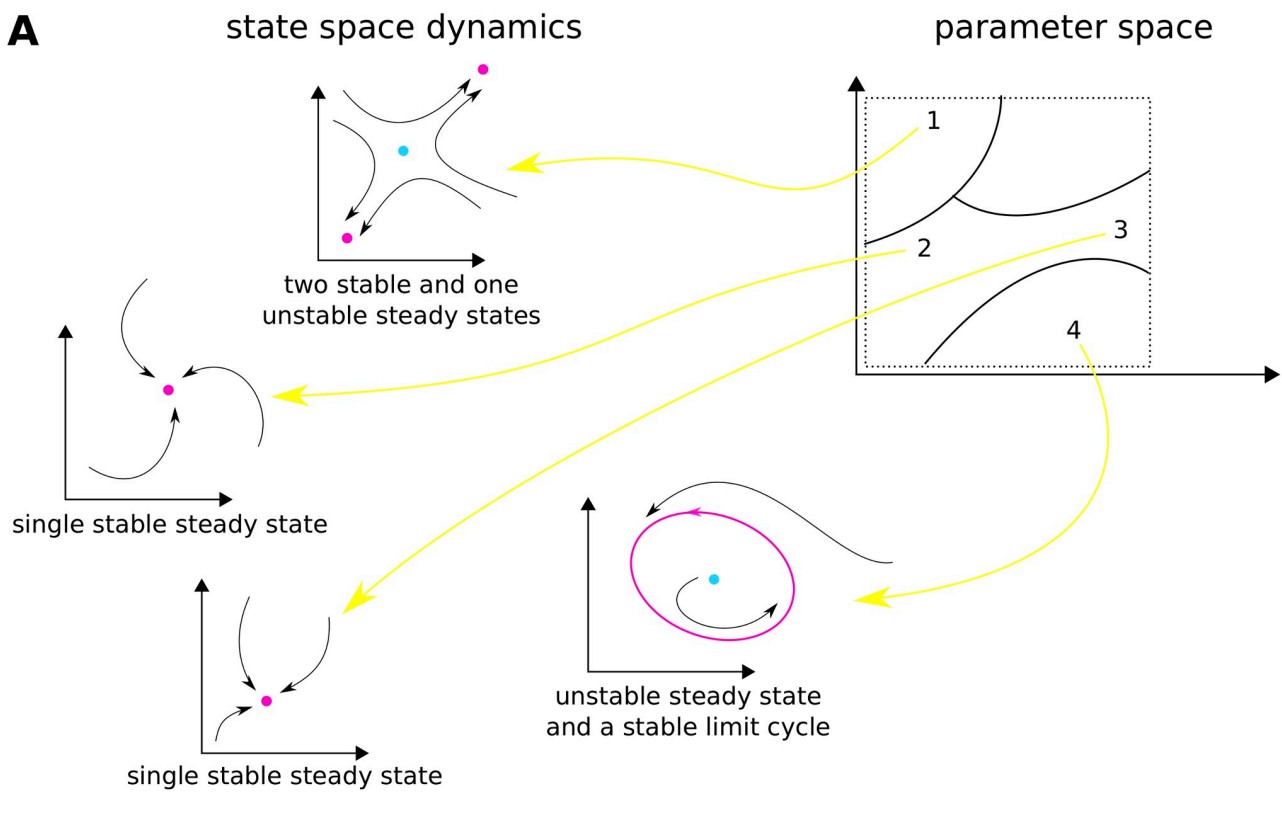

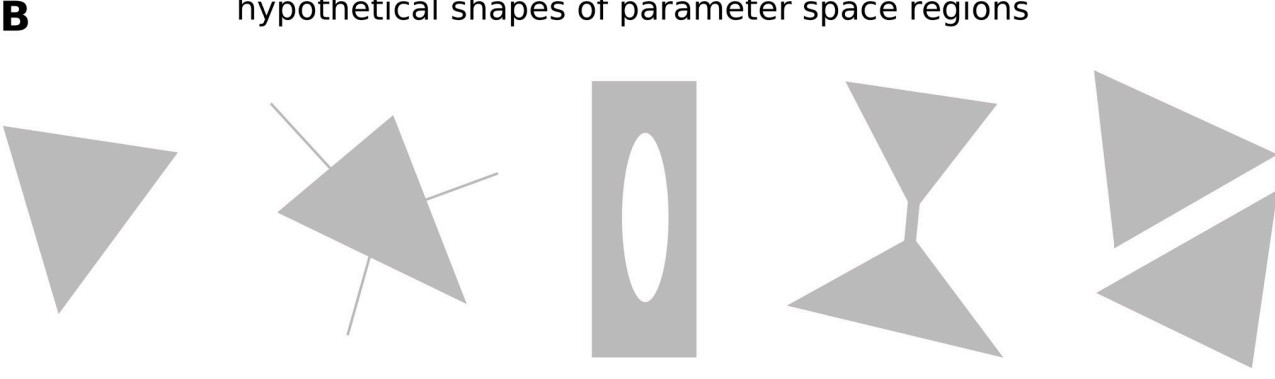

**Fig 1. Models and their parameter geography.** A: Behaviour of a hypothetical mathematical model, as in a system of ordinary differential equations. When a point is chosen in the parameter space (right), indicated by a number 1, . . ., 4, a dynamics takes place in the state space (left), shown by the trajectories with arrowheads. The state space and parameter space are shown here as 2-dimensional but could be of any dimension. The parameter space is expected to break up into regions, shown here within a box of finite volume (dotted boundary) following the method adopted in the paper, such that the dynamics remains qualitatively similar within each region, as for parameter points 2 and 3, and becomes qualitatively different between regions. Parameter point 1 gives bistability in the dynamics, parameter points 2 and 3 give monostability, and parameter point 4 gives a stable limit cycle. Stable attractors are magenta; unstable attractors are cyan. B: Hypothetical shapes of regions in parameter space, assumed to be within a finite volume in two dimensions. All except region 2 have nonzero volume in two dimensions in the vicinity of each point. Region 1 is convex; region 2 has one-dimensional subregions, which would not be detectable by random sampling in two dimensions; region 3 has a hole in its interior; region 4 has a narrow "neck;" region 5 is disconnected.

Bifurcations arise on the boundaries between regions and give rise to the abrupt change in qualitative dynamics from one parameter region to the next.

System properties whose robustness is being assessed are typically defined for particular dynamical portraits and are therefore properties of one or more parametric regions. We will focus here on the property of bistability: the existence in the dynamical portrait of two stable steady-states, accompanied by one unstable steady-state, as in parameter point 1 in Fig 1A. Technically speaking, we will work in terms of stationarity—the existence of steady-states—rather than stability, which requires an assessment of local dynamics. However, as a convenience of language, we will continue to use "monostability" and "bistability" in favour of the less euphonious "monostationarity" and "tristationarity," respectively, and we explain further below the issues involved in distinguishing between these properties.

Bistability has been widely interpreted as the mathematical counterpart of biological decision making, switching, or memory. It has been used, for example, to interpret state switching in single cells, both in unicellular organisms [12, 13] and in individual cells within multi-cellular organisms [14–18]; state switching in whole organs [19]; cell lineage choice during organismal development [20–25], where the implications of bistability have been widely reviewed [26–28]; and memory formation during signal transduction [14, 29, 30] and neuronal learning [31–33].

Here, we will consider bistability arising from protein post-translational modification (PTM). PTM is the mechanism by which amino acid residues in a protein are covalently modified in response to physiological conditions, through the catalytic action of forward-modifying and reverse-demodifying enzymes [34]. Phosphorylation is the most widely studied modification, but many others are now known and PTM is a central mechanism in cellular information processing [35]. Models show that bistability, and even multistability with more than two stable steady-states, can emerge in PTM systems, provided a substrate protein is modified on two or more sites by one forward and one reverse enzyme [29, 30]. Such multisite modification is common, and bistability in PTM has been suggested as the basis for cellular memory [36].

The choice of PTM as a bistability mechanism has the advantage that steady-states can be realised as solutions to polynomial equations. This permits analysis by numerical solution of polynomial equations rather than by numerical integration of differential equations. The former is much faster computationally. This allows the robustness problem to be addressed by randomly generating points in parameter space and identifying those which give rise to bistability. In this way, the bistable region can be effectively characterised. Such a statistical approach has interesting parallels with high-dimensional data analysis, although, here, the data arise not from experiments but from a model.

Many kinds of approaches have been taken to quantitatively assess robustness in this way, such as by parametric sensitivity [37–40], or by estimating volume and shape [41–47]. Algebraic methods can sometimes provide an analytical description of parametric regions [46, 48–51], but these methods tend to scale poorly with the complexity of the system. For systems arising from networks of biochemical reactions, methods also exist which give parametric conditions under which bistability occurs [52–58], or does not occur [59–62], and some of these apply to PTM systems [54, 57, 58, 63–66]. Discriminant locus approaches based on more general algebraic geometric techniques have been used to find regions of parameter space for which appropriate generic behaviours occur [67]. Bistable parametric regions have thereby been demarcated in various contexts [49, 53, 56, 58, 64, 66]. However, the relevant conditions for bistability are typically sufficient, but not always necessary, making it difficult in some cases to exactly determine bistable regions. Furthermore, these kinds of results also typically require a complete description of the underlying network, which makes it difficult to rise above the biochemical complexity.

Here, we build upon the approach of exploring the size and shape of parametric regions. Such "parameter geography" seeks to make a global assessment of the bistable region. The first property to consider is the dimension of the region. If the parameter space has dimension $m$, the bistable region may also be of dimension $m$ (Fig 1B, example 1), of lower dimension, or some combination of the two (Fig 1B, example 2). Lower dimensionality has nearly always been neglected in the biological literature because it would never be found by random sampling. However, we are not aware of theorems that would rule it out for a general dynamical system, and it could conceivably arise from some mathematical constraint or degeneracy among the parameters. Therefore, to be careful, results obtained by parametric sampling should be qualified by the statement "with probability one," to allow for any subsets of lower local dimension that are invisible to the sampling process. We will take this caveat for granted in what follows.

Assuming the bistable region has full dimension relative to the ambient parameter space, so that points within it can be found by sampling, a concise, global measure of robustness is the $m$-dimensional volume of the region, as contained within some $m$-dimensional box of finite extent (Fig 1A). This may be statistically estimated by counting the proportion of points in the region. The larger the volume, the more bistable parameter points and the more robust the property of bistability. However, volume gives no information about the region's shape [46], which may conceivably exhibit local features, such as interior holes and cavities (Fig 1B, example 3) or waists (Fig 1B, example 4). Relatively small changes to parameter values in such local regions may destroy bistability and compromise robustness. Convexity offers a test for this. A region is convex if, given any two points within the region, the straight line segment connecting the two points also lies within the region (Fig 1B, example 1). Convexity is a strong property of a region but a more nuanced "visibility ratio" can be estimated by randomly choosing pairs of points from the region and estimating the frequency with which the line segment between each pair lies entirely in the region. The higher the visibility ratio, the closer the region is to convexity and the more robust the property of bistability. Another measure of shape is topological connectedness. A region is disconnected if it consists of two or more separated pieces (Fig 1B, example 5). Lack of connectivity may indicate that bistability arises for different reasons, which may have different degrees of robustness. Connectedness may be estimated using a connectivity graph, originally developed for robotic motion planning. These measures of volume, convexity, and connectedness will be the focus of the results presented here, but we note that they are only a first step towards understanding the complexities of shape in high dimensions [68].

Two recent developments, one mathematical and one computational, make the random sampling of high-dimensional parameter space feasible for determining parameter geography under realistic biochemical assumptions. We briefly describe the two developments here, with further details in the main text.

First, we use the graph-based linear framework for timescale separation to describe PTM systems [69, 70]. The framework offers several advantages. To begin with, it allows the enzyme mechanism underlying each modification to be treated in a general and realistic manner, instead of having to assume only the Michaelis-Menten reaction scheme. Specifically, an enzyme $E$ that converts substrate $S$ into product $P$ can follow any mechanism that is built up from the elementary reactions in the following "grammar,"

$$S + E \rightarrow Y_i \, , \qquad Y_j \rightarrow Y_k \, , \qquad Y_\ell \rightarrow P + E \, , \qquad (1)$$

where the $Y$'s are intermediate enzyme-substrate complexes [71, 72]. This allows a mechanism to take multiple routes with multiple intermediates and to be <u>irreversible</u> (product cannot be

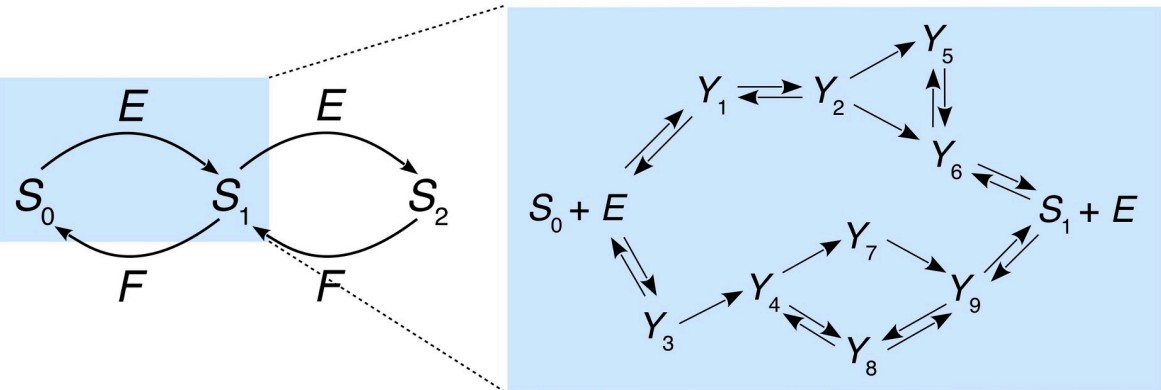

**Fig 2. Reaction network and example enzyme mechanism for a two-site PTM system.** A two-site PTM system is shown, in which modification and demodification are sequential and in which each enzymatic step yields a single product ("distributivity"). The box on the right shows an example of an enzyme mechanism made up from the elementary reactions in the grammar in Eq 1, illustrating multiple routes and multiple intermediate enzyme-substrate complexes. This example is weakly irreversible: the product $S_1$ can bind to $E$ but cannot be converted back into substrate $S_0$, so that the mechanism is irreversible overall.

converted back into substrate) without being <u>strongly irreversible</u> (product does not rebind to enzyme). Fig 2 shows a linear-framework graph using the reactions (edges) in Eq 1 for an example of a <u>weakly irreversible</u> mechanism, i.e., a mechanism in which product is not converted to substrate but can rebind to enzyme.

The significance of weak irreversibility is frequently overlooked. Forward modification and reverse demodification of a protein may well be effectively irreversible under physiological conditions but this does not imply absence of product rebinding. If the concentration of product is appreciable, as will often be the case in a PTM system, then the product must be expected to rebind to the enzyme that produced it. Indeed, it is a requirement of thermodynamics that binding and unbinding events, which draw their energy from the surrounding thermal bath, must be reversible [70]. Any strongly irreversible mechanism, such as the Michaelis-Menten scheme, fails to satisfy this requirement. (We note, however, that it was a perfectly appropriate assumption for Michaelis and Menten [73].) Despite such difficulties being repeatedly pointed out [70, 74, 75], the Michaelis-Menten scheme remains almost universally used for describing enzyme kinetics. We were particularly interested, therefore, in understanding how the different irreversibility assumptions would influence parameter geography and the assessment of robustness.

Regardless of the complexity of the reaction mechanism built from the grammar in Eq 1, the steady-state behaviour of the mechanism can be summarised with just four generalised parameters, two for the forward direction and two for the reverse direction [71, 72]. These parameters can be thought of as versions of the catalytic efficiency and Michaelis-Menten constant for the simple Michaelis-Menten scheme. By using these generalised parameters, in place of the many individual rate constants for each mechanism, it becomes possible to make general statements about steady-state behaviour for systems in which each enzyme follows its own reaction mechanism subscribing to the grammar in Eq 1 [72]. Due attention can thereby be paid to the behaviour of individual enzymes, which are known to exhibit many different kinds of reaction mechanisms [76]. Note, in particular, that our results, although obtained numerically, are valid for an infinite class of models, corresponding to different choices of mechanisms from the grammar in Eq 1.

For a PTM system, the linear framework further allows the exponential combinatorial complexity arising from multiple modification sites to be eliminated at steady-state [17, 30]. The steady-state behaviour of any PTM system can be reduced in this way to the solution of $k$ polynomial equations in $k$ variables, where $k$ is the number of enzymes in the system. The number of modification sites influences the degrees of these equations but not the number of variables. For the case of a two-site PTM system with one forward and one reverse enzyme, this elimination procedure yields two polynomial equations, each of total degree 4 in two variables (Eq 10). These equations have eight non-dimensional parameters, which are defined in terms of the generalised parameters for the two enzymes, and three conserved quantities, which correspond to the total amounts of substrate and enzymes.

The variables in the polynomial equations are the normalised steady-state concentrations of the (free) enzymes, from which the steady-state concentrations of all other components in the PTM system can be determined. Solutions of the polynomial equations correspond exactly to the steady-states of the PTM system. Numerical integration of the underlying differential equations is thereby avoided. The linear framework allows us to rise above the details of enzyme mechanisms and the combinatorial complexity of PTM, at least for describing the steady-state behaviour [70].

The second development on which we rely are advances in numerical algebraic geometry for solving polynomial equations, implemented in the software tools, Bertini, Paramotopy, and alphaCertified [77, 78]. Algebraic geometry deals with the mathematical structures that arise as solutions to polynomial equations and has already been applied to systems biology [48, 49]. Bertini numerically solves polynomial equations by "homotopy continuation:" it starts from a system of polynomial equations whose solutions are known, then continuously deforms these solutions through a homotopy until they coincide, up to arbitrary numerical precision, with the solutions of the system of interest. The solutions along the homotopy are tracked using predictor-corrector methods. Paramotopy extends this procedure to efficiently track homotopies in parameter space, thereby facilitating the parallel solution of a system of parameterised polynomial equations at many different parameter values. Finally, alphaCertified can be used to rigorously determine whether each approximate numerical solution found by Bertini lies near a true solution to the equations, and thus confirm the accuracy of our calculations [78].

In summary, the linear framework enables model reduction of a realistic PTM system to two polynomial equations, while Bertini, Paramotopy and alphaCertified enable efficient and accurate solution of these equations. Their combination allows us to determine the steady-state behaviour of the two-site PTM system at a total of $\sim 10^9$ parameter points in five different hypercubes in both an 8-dimensional parameter space for weak irreversibility and a 6-dimensional parameter space for strong irreversibility. We thereby map the parameter geography of bistability, from which several interesting and unexpected conclusions emerge.

We find that the bistable volume increases, in an apparently monotonic and sigmoidal manner, as the substrate grows more abundant relative to the enzymes, and there is a threshold substrate level below which bistability is undetectable by random sampling. Strikingly, we find that the bistable region occupies a much smaller proportion of the sampling domain under weak irreversibility than under strong irreversibility, and we demonstrate a tradeoff between bistability and product rebinding that underlies this discrepancy. We also find that, despite the apparently monotonic growth in the bistable volume, the region itself does not grow monotonically: parameter points can move back and forth between monostability and bistability. We formulate these observations as mathematical conjectures and questions that invite further analysis.

## Results

### Steady-state polynomial equations

We give an overview here of how the steady-state polynomial equations are derived, focusing on the generalised parameters and the process of model reduction, as described in the Introduction. Full details of the calculation are provided in the Materials and Methods.

We consider a protein, $S$, that is post-translationally modified at two sites by a forward-modifying enzyme, $E$, and a reverse-demodifying enzyme, $F$ (Fig 2). We assume that modification takes place in a specific site order and that demodification takes place in the reverse order, so that there are only three modification states, or "modforms" [34]. The modforms will be denoted by $S_i$, where $i$ is the number of modified sites. These assumptions reduce the algebraic complexity of the equations, thereby permitting more extensive parametric exploration, but the methods presented here may be applied more generally.

We assume that $E$ and $F$ follow any reasonable distributive reaction mechanism built up from the grammar in Eq 1. Here, "reasonable" means only that the mechanism should be able to convert substrate to product and not yield only a dead-end complex; see [72] for details. A distributive ("hit-and-run") reaction is one that yields only a single product with a given substrate. Processive ("bind-and-slide") reactions, in which the enzyme catalyses multiple modifications while remaining bound to the substrate, can also be accommodated within the grammar, but can yield more complex behaviours [79, 80]. Each enzyme has two substrates—$S_0$ and $S_1$ for $E$, and $S_1$ and $S_2$ for $F$—and may use a different mechanism from the grammar on each substrate.

The linear framework shows that the steady-state behaviour of each reaction mechanism can be summarised with just four generalised parameters. For the case of $E$ converting $S_0$ to $S_1$, which we will denote by the shorthand $S_0 \xrightarrow{E} S_1$, there are two reciprocal total generalised Michaelis-Menten constants (rtgMMCs), $\kappa_{0,1}^E$ and $\kappa_{1,0}^E$, and two total generalised catalytic efficiencies (tgCEs), $c_{0,1}^E$ and $c_{1,0}^E$. One parameter of each pair follows the forward direction in which $S_0$ is converted to $S_1$, indicated by the subscript "0, 1", while the other parameter follows the reverse direction in which $S_1$ is converted to $S_0$, indicated by the subscript "1, 0".

The rtgMMCs, $\kappa_{0,1}^E$ and $\kappa_{1,0}^E$, respectively determine the extent to which $S_0$ and $S_1$ bind to $E$ to form the intermediate complexes in the reaction mechanism:

$$\kappa_{0,1}^E [E][S_0] + \kappa_{1,0}^E [E][S_1] = \sum_{S_0 \xrightarrow{E} S_1} [Y_*] . \tag{2}$$

Here, and in what follows in the rest of the paper, $[X]$ denotes the steady-state concentration of $X$, and $Y_*$ is a shorthand for those intermediate complexes appearing in the reaction mechanism given in the subscript of the summation. This avoids having to introduce notation for the individual intermediates when these details are not necessary. The rtgMMCs have units of (concentration)$^{-1}$. The parameter $\kappa_{1,0}^E$ measures the extent to which the product of the reaction, $S_1$, can bind to $E$, thereby sequestering the enzyme from its substrate, $S_0$, and giving rise to product inhibition [81].

The tgCEs, $c_{0,1}^E$ and $c_{1,0}^E$, determine the rate at which $E$ converts $S_0$ to $S_1$ and the rate at which $E$ converts $S_1$ to $S_0$, respectively. The reaction $S_0 \xrightarrow{E} S_1$ incurs the following rate contributions:

$$\frac{d}{dt}[S_0] = \cdots + c_{1,0}^E [E][S_1] - c_{0,1}^E [E][S_0] + \cdots$$

$$\frac{d}{dt}[S_1] = \cdots + c_{0,1}^E [E][S_0] - c_{1,0}^E [E][S_1] + \cdots , \tag{3}$$

where the dots indicate similar rate contributions from the other three reactions (Eq 22). The tgCEs have units of (concentration $\cdot$ time)$^{-1}$.

The generalised parameters are given by rational expressions in the rate constants of the corresponding reaction mechanisms. These expressions can be explicitly described once these mechanisms are specified in the grammar of Eq 1 [72]. Different mechanisms yield different expressions for the generalised parameters, but the steady-state behaviour of the mechanism is independent of the details of these expressions.

Modification and demodification in PTM systems are energy-dissipating and regarded as irreversible under physiological conditions [35]. We therefore assume that the enzymes operate irreversibly, so that, using $S_0 \xrightarrow{E} S_1$ as an example,

$$\kappa_{0,1}^E > 0, \qquad c_{0,1}^E > 0, \qquad \kappa_{1,0}^E \geq 0, \qquad c_{1,0}^E = 0. \tag{4}$$

This ensures positive flux of substrate $S_0$ into product $S_1$ ($c_{0,1}^E > 0$), which also requires binding of substrate to enzyme ($\kappa_{0,1}^E > 0$), but no flux of product into substrate ($c_{1,0}^E = 0$), so that the reaction is irreversible overall. Product rebinding is permitted ($\kappa_{1,0}^E \geq 0$) and strong irreversibility arises when $\kappa_{1,0}^E = 0$. Weak irreversibility corresponds to $\kappa_{1,0}^E > 0$.

The PTM system has four separate reactions, each of which has three nonzero generalised parameters, giving 12 parameters in all. In reducing the system to two polynomial equations in two variables, the number of parameters is further reduced from 12 to 8. We briefly summarise here the three key steps in the model reduction, leaving full details to the Materials and Methods.

The first step arises from the steady-state assumption. Because modification and demodification are assumed to be ordered (Fig 2), the net flux through each modification loop must be zero [82]. Hence, using Eqs 3 and 4 (see also Eq 22),

$$c_{0,1}^E [E][S_0] = c_{1,0}^F [F][S_1] \qquad \text{and} \qquad c_{1,2}^E [E][S_1] = c_{2,1}^F [F][S_2].$$

Hence, $[S_1]$ and $[S_2]$ can be determined in terms of $[S_0]$, $[E]$, and $[F]$, as

$$[S_1] = \alpha \left( \frac{[E]}{[F]} \right) [S_0] \qquad \text{and} \qquad [S_2] = \beta \left( \frac{[E]}{[F]} \right) [S_1], \tag{5}$$

where

$$\alpha = \frac{c_{0,1}^E}{c_{1,0}^F} \qquad \text{and} \qquad \beta = \frac{c_{1,2}^E}{c_{2,1}^F} \tag{6}$$

are new non-dimensional parameters. This reduces the number of parameters from 12 to 10.

The second step arises from conservation of the substrate, which leads to the equation,

$$S_{\text{tot}} = [S_0] + [S_1] + [S_2] + \sum_{S_0 \xrightarrow{E} S_1} [Y_*] + \sum_{S_1 \xrightarrow{F} S_0} [Y_*] + \sum_{S_1 \xrightarrow{E} S_2} [Y_*] + \sum_{S_2 \xrightarrow{F} S_1} [Y_*], \tag{7}$$

for some positive constant $S_{\text{tot}}$. Using Eqs 2 and 5, this allows $[S_0]$, and therefore $[S_1]$ and $[S_2]$, to be determined in terms of $[E]$ and $[F]$. All the state variables have now been eliminated in favour of $[E]$ and $[F]$. (See Eqs 23–25).

The third and final step arises from the conservation of the enzymes, which leads to the two equations,

$$E_{\text{tot}} = [E] + \sum_{S_0 \xrightarrow{E} S_1} [Y_*] + \sum_{S_1 \xrightarrow{E} S_2} [Y_*]$$

$$\tag{8}$$

$$F_{\text{tot}} = [F] + \sum_{S_1 \xrightarrow{F} S_0} [Y_*] + \sum_{S_2 \xrightarrow{F} S_1} [Y_*],$$

for positive constants $E_{\text{tot}}$ and $F_{\text{tot}}$. Using Eq 2 and the expressions for $[S_0]$, $[S_1]$, and $[S_2]$ described above, this yields two equations for $[E]$ and $[F]$, which fully determine the steady-state. The remaining state variables can be expressed as rational functions of the steady-state values of $[E]$ and $[F]$. (See Eqs 26 and 27).

This result is a particular instance of the general theorem that, if a PTM system has $k$ enzymes operating on a single substrate, then, irrespective of the number of sites and the mechanisms of the enzymes, the steady-state of each state variable is a rational function of the $k$ steady-state enzyme concentrations, and these concentrations can be obtained as the solutions to a system of $k$ equations in $k$ unknowns [17]. Here, $k = 2$.

The three conserved totals, $S_{\text{tot}}$, $E_{\text{tot}}$, and $F_{\text{tot}}$, are different in character from the parameters of the system because they are determined by the initial conditions. In the biological interpretation, these conserved totals can be modulated by changes in physiological conditions. We therefore seek to understand the parameter geography of the system as these totals are varied.

The elimination process above requires only the composite parameters $\kappa_1^E = \kappa_{1,0}^E + \kappa_{1,2}^E$ and $\kappa_1^F = \kappa_{1,0}^F + \kappa_{1,2}^F$ (Materials and methods), which summarise the binding of the intermediate modform, $S_1$, to $E$ and $F$, respectively. This further reduces the number of parameters from 10 to 8.

It is always more convenient to work with non-dimensional parameters, and $\alpha$ and $\beta$ are already non-dimensional. The other six parameters involve the rtgMMCs and we choose to non-dimensionalise them using the corresponding enzyme totals,

$$\epsilon_0 = \kappa_{0,1}^E E_{\text{tot}} > 0, \qquad \epsilon_1 = \kappa_1^E E_{\text{tot}} > 0, \qquad \epsilon_2 = \kappa_{2,1}^E E_{\text{tot}} \geq 0,$$

$$\tag{9}$$

$$\phi_0 = \kappa_{0,1}^F F_{\text{tot}} \geq 0, \qquad \phi_1 = \kappa_1^F F_{\text{tot}} > 0, \qquad \phi_2 = \kappa_{2,1}^F F_{\text{tot}} > 0.$$

The $\epsilon$'s summarise the binding characteristics of the reactions catalysed by $E$; the $\phi$'s summarise the binding characteristics of the reactions catalysed by $F$; and $\alpha$ and $\beta$ are ratios that compare the catalytic efficiencies of the reactions catalysed by $E$ and $F$.

The constraints on the parameters in Eq 9 arise from an interesting asymmetry between the reactions in which $S_1$ is a product, $S_0 \xrightarrow{E} S_1$ and $S_2 \xrightarrow{F} S_1$, and the reactions in which $S_1$ is a substrate, $S_1 \xrightarrow{E} S_2$ and $S_1 \xrightarrow{F} S_0$. Strong irreversibility of the latter reactions influences the parameters in Eq 9: $\epsilon_2 = 0$ if, and only if, $S_1 \xrightarrow{E} S_2$ is strongly irreversible; and $\phi_0 = 0$ if, and only if, $S_1 \xrightarrow{F} S_0$ is strongly irreversible. However, strong irreversibility of the former reactions has no such effect: if $S_0 \xrightarrow{E} S_1$ is strongly irreversible, so that $\kappa_{1,0}^E = 0$, it is still the case that $\kappa_{1,2}^E > 0$ (Eq 4), so that $\kappa_1^E > 0$. In other words, even if $S_1$ is unable to sequester $E$ as the product of the reaction $S_0 \xrightarrow{E} S_1$, it is still able to bind to $E$ by being the substrate of the reaction $S_1 \xrightarrow{E} S_2$. Similarly, $\kappa_1^F > 0$, irrespective of whether or not the reaction $S_2 \xrightarrow{F} S_1$ is strongly irreversible. It follows that the four parameters, $\epsilon_0$, $\epsilon_1$, $\phi_1$ and $\phi_2$, in Eq 9 are always positive, while the remaining two parameters, $\epsilon_2$ and $\phi_0$, are non-negative.

The three conserved totals can be non-dimensionalised as follows,

$$\sigma = \frac{S_{\text{tot}}}{E_{\text{tot}}} \, , \qquad \lambda = \frac{S_{\text{tot}}}{F_{\text{tot}}} \, , \qquad \zeta = \frac{E_{\text{tot}}}{F_{\text{tot}}} \, .$$

Finally, the two state variables can be non-dimensionalised using the corresponding enzyme totals,

$$u = \frac{[E]}{E_{\text{tot}}} \, , \qquad v = \frac{[F]}{F_{\text{tot}}} \, .$$

Non-dimensionalisation can be performed in different ways, which can lead to different insights; the method adopted here works well for this particular analysis. The non-dimensional parameters and non-dimensional totals are all assumed to be positive, except when strong irreversibility is imposed on $S_1 \xrightarrow{E} S_2$ or $S_1 \xrightarrow{F} S_0$, in which case $\epsilon_2 = 0$ or $\phi_0 = 0$, respectively. The other parameters and variables are always taken to be positive.

The Materials and Methods show that, once the dust of calculation has settled, we arrive at the equations $\Phi_1(u, v) = 0$ and $\Phi_2(u, v) = 0$, where

$$\Phi_1(u, v) =$$

$$\alpha v^2 + \zeta u v + \beta \zeta^2 u^2 + (\alpha \epsilon_0 - \alpha \epsilon_0 \sigma - \alpha + \phi_1 \zeta) u v^2$$

$$+ (\epsilon_1 \zeta - \epsilon_1 \zeta \sigma - \zeta + \beta \phi_2 \zeta^2) u^2 v + (\beta \epsilon_2 \zeta^2 - \beta \epsilon_2 \zeta^2 \sigma - \beta \zeta^2) u^3 + \alpha \phi_0 v^3$$

$$- (\alpha \epsilon_0 + \phi_1 \zeta) u^2 v^2 - (\epsilon_1 \zeta + \beta \phi_2 \zeta^2) u^3 v - \beta \epsilon_2 \zeta^2 u^4 - \alpha \phi_0 u v^3$$

$$\Phi_2(u, v) =$$

$$\alpha v^2 + \zeta u v + \beta \zeta^2 u^2 + (\alpha \phi_0 - \alpha \phi_0 \lambda - \alpha) v^3 + (\phi_1 \zeta - \phi_1 \zeta \lambda - \zeta + \alpha \epsilon_0) u v^2$$

$$+ (\beta \phi_2 \zeta^2 - \beta \phi_2 \zeta^2 \lambda - \beta \zeta^2 + \epsilon_1 \zeta) u^2 v + \beta \epsilon_2 \zeta^2 u^3 - (\alpha \epsilon_0 + \phi_1 \zeta) u v^3$$

$$- (\epsilon_1 \zeta + \beta \phi_2 \zeta^2) u^2 v^2 - \beta \epsilon_2 \zeta^2 u^3 v - \alpha \phi_0 v^4 \, .$$

(10)

Here, $\Phi_1$ and $\Phi_2$ are each polynomial of total degree 4 in the non-dimensional variables $u$ and $v$, with eight non-dimensional parameters and three non-dimensional totals. The polynomial equations in Eq 10 will be the object of analysis in the rest of the paper.

## General approach to parameter geography

We describe here the general approach we take to exploring the parameter geography of the bistable region, which is then used in all subsequent sections of the paper. To keep the analysis relatively simple, we assume that $E_{\text{tot}} = F_{\text{tot}}$, so that $\zeta = 1$ and $\sigma = \lambda$, and take $\sigma$ to be the parameter that varies. If $\sigma > 1$, so that $S_{\text{tot}} > E_{\text{tot}} = F_{\text{tot}}$, then both enzymes approach saturation by the substrate, which is known to promote bistability. We therefore began our analysis by examining parameter geography for 15 values of $\sigma$,

$$\sigma = 1.0 \, , \; 1.5 \, , \; 2.0 \, , \; 2.5 \, , \; 3.0 \, , \; 4.0 \, , \; 5.0 \, , \; 7.0 \, , \; 10 \, , \; 15 \, , \; 20 \, , \; 50 \, , \; 100 \, , \; 200 \, , \; 500 \, . \quad (11)$$

We order the eight non-dimensional parameters so that, if $\theta \in \mathbb{R}^8$ is a point in parameter space, then

$$\theta_1 = \alpha \, , \;\; \theta_2 = \beta \, , \;\; \theta_3 = \epsilon_0 \, , \;\; \theta_4 = \epsilon_1 \, , \;\; \theta_5 = \epsilon_2 \, , \;\; \theta_6 = \phi_0 \, , \;\; \theta_7 = \phi_1 \, , \;\; \theta_8 = \phi_2 \, .$$

Throughout most of the analysis which follows, we consider a finite-volume box in parameter

space, $\mathcal{H} = [0.1, 10]^8 \subset \mathbb{R}^8$, which constrains each parameter to lie in the interval [0.1, 10]. Each non-dimensional parameter is therefore positive. As previously noted, this means that the reactions in which $S_1$ is a substrate, $S_1 \xrightarrow{E} S_2$ and $S_1 \xrightarrow{F} S_0$, are weakly irreversible, although the reactions in which $S_1$ is a product, $S_0 \xrightarrow{E} S_1$ and $S_2 \xrightarrow{F} S_1$, may be either strongly or weakly irreversible. Weak irreversibility is the physically realistic assumption and we focus on that first.

The range [0.1, 10] sets the nominal value of each non-dimensional parameter to 1. This is appropriate for $\theta_1 = \alpha$ and $\theta_2 = \beta$ because they are ratios of tgCEs for $E$ and $F$ (Eq 6). The other non-dimensional parameters, however, are products of rtgMMCs and conserved totals (Eq 9) and their nominal values are harder to judge. While some experimental data are available, estimated values can vary widely. In the absence of broadly acknowledged values, we chose 1 as the nominal value for all non-dimensional parameters.

Bézout's Theorem from algebraic geometry tells us that the typical number of solutions of a system of polynomial equations is given by the product of the total degrees of the polynomials [83]. This gives 16 solutions for the two equations in Eq 10, which each have total degree 4. However, "solution" has to be interpreted carefully. Bézout's Theorem holds over the field of complex numbers. The equation $x^2 + 1 = 0$ has two complex solutions, $x = \pm i$, but no real solutions. Bézout's Theorem also requires the use of projective space, which allows solutions at infinity, like that of the equations $x + y = 1$ and $x + y = 2$, which do not intersect in Euclidean space. Finally, solutions may be repeated, as in the case of the equation $(x - 1)^2 = 0$, in which case they must be counted with the appropriate multiplicity.

In practice, we found that, given $\zeta = 1$, a fixed value of $\sigma$, and a randomly chosen point $\theta \in \mathcal{H}$, the software tool Bertini yields the following 16 complex solutions for Eq 10: the zero solution, $u = v = 0$, which is always a solution of Eq 10, has multiplicity 6; seven additional finite solutions; and three solutions which are projectively at infinity. This pattern of solutions is generic: departures from the pattern can only occur on a subset of probability zero (Lebesgue measure zero) in $\mathbb{R}^8$ [83]. Accordingly, departures do not occur for randomly selected parameter points in $\mathbb{R}^8$. However, genericity can sometimes be lost during numerical homotopy continuation in Bertini. We developed a systematic procedure for addressing this (Materials and methods), which may be of interest in other studies. We also used the software tool alphaCertified to confirm that representative random samples of our numerical solutions were in the vicinity of true solutions, thereby greatly reducing the possibility of numerical artifacts (Materials and methods). Importantly, of the seven finite, nonzero solutions of Eq 10 at each parameter point, we always found either one or three positive real solutions.

Throughout this analysis, we refer to the occurrence of one positive real solution as monostability, and the occurrence of three positive real solutions as bistability (assuming two stable steady-states and one unstable one). We use this language only as a convenience and note the importance of distinguishing between stationarity and stability. For example, recent work on "mixed-mechanism" PTM systems, which incorporate both distributive and processive mechanisms, has shown the existence of a single unstable steady-state with limit-cycle oscillations [79, 80]. Such behaviours are not known to be a feature of PTM systems that employ only distributive mechanisms but checking for them requires testing for stability. This is not straightforward within the algebraic approach taken here. Eq 10 does not provide information about the stability of its solutions, which depends on the transient behaviour of the system near a steady-state. To determine stability, it is necessary to fix the mechanism of each enzyme, as built up from the grammar in Eq 1, and analyse the corresponding system of differential equations. If the steady-state is hyperbolic, asymptotic stability can be determined from the eigenvalues of the Jacobian matrix [11]. However, this would leave open the question of whether the same stability would be found for other choices of enzyme mechanism. We decided, therefore,

to set aside the stability question and to focus on what can be deduced algebraically about steady-states from Eq 10. With that in mind, as mentioned above and in the Introduction, we use the terms "monostability" and "bistability" only for convenience, in place of "monostationarity" and "tristationarity," respectively.

## Bistable volume increases sigmoidally with $\sigma$

Having explained our general approach in the previous section, we begin the analysis by introducing the parametric region of interest. Recall that $\mathcal{H} = [0.1, 10]^8 \subset \mathbb{R}^8$ is the box in parameter space in which we will work. Given $\zeta = 1$ and a value of $\sigma$, let $\mathcal{M}_\sigma \subset \mathcal{H}$ be the subset of parameter points in $\mathcal{H}$ at which the system is bistable,

$$\mathcal{M}_\sigma = \{\theta \in \mathcal{H} \mid \exists \text{ three positive real solutions to Eq. 10}\}.$$

Our main goal in the paper is to explore how the size and shape of $\mathcal{M}_\sigma$ changes as $\sigma$ takes the values listed in Eq 11. We start our exploration with the volume of the bistable region.

Let $\iota_\sigma : \mathcal{H} \to \{0, 1\}$ be the indicator function of $\mathcal{M}_\sigma$:

$$\iota_\sigma(\theta) = \begin{cases} 1 & \text{if } \theta \in \mathcal{M}_\sigma \\ 0 & \text{otherwise}. \end{cases}$$

Let $V_\mathcal{H} = 2^8 = 256$ denote the (base ten) logarithmic volume of $\mathcal{H}$, and let $V_\sigma$ denote the logarithmic volume of $\mathcal{M}_\sigma$, normalised to that of $\mathcal{H}$. This is conveniently defined by the integral of the indicator function,

$$V_\sigma = \frac{1}{V_\mathcal{H}} \int_{\theta \in \mathcal{H}} \iota_\sigma(\theta) \, d \log \theta. \tag{12}$$

The integral in Eq 12 is taken with respect to the logarithmic measure on $\mathbb{R}^8$. It cannot be evaluated analytically but lends itself to efficient unbiased statistical estimation by Monte Carlo methods, in which parameter points in $\mathcal{H}$ are randomly sampled. Specifically, $V_\sigma$ can be approximated as the proportion of randomly sampled points in $\mathcal{H}$ that lie in $\mathcal{M}_\sigma$, where "random" means with respect to the logarithmic measure. This amounts to independently sampling the logarithm of each parameter, $\log \theta_i$, from the uniform distribution on $[-1, 1]$. We refer to this as ILR (Independent Logarithmic Random) sampling.

Given a set $\mathcal{S}$ of parameter points, randomly chosen in this way, an unbiased statistical estimator for $V_\sigma$ is given by

$$\widehat{V}_\sigma = \frac{1}{\#\mathcal{S}} \sum_{\theta \in \mathcal{S}} \iota_\sigma(\theta), \tag{13}$$

where $X$ denotes the number of elements in the finite set $X$. Confidence intervals for the estimator in Eq 13 may be computed using the central limit theorem (Materials and methods).

To perform this estimation, we ran Bertini and Paramotopy on a large computing cluster (Materials and methods, S1 Appendix). We generated an ILR sample of $4 \times 10^6$ points in $\mathcal{H}$ to calculate $\widehat{V}_\sigma$ for all values of $\sigma$ in Eq 11 greater than 2.5, for each of which we found sufficiently many bistable points to achieve good statistical confidence. For $\sigma = 2.5$, we generated an additional ILR sample of $2 \times 10^6$ points in $\mathcal{H}$ and calculated $\widehat{V}_\sigma$ using the combined sample of $6 \times 10^6$ points; and, for $\sigma = 1.0, 1.5, 2.0$, we generated a third ILR sample of $4 \times 10^6$ points in $\mathcal{H}$ and calculated $\widehat{V}_\sigma$ using the combined sample of $10^7$ points.

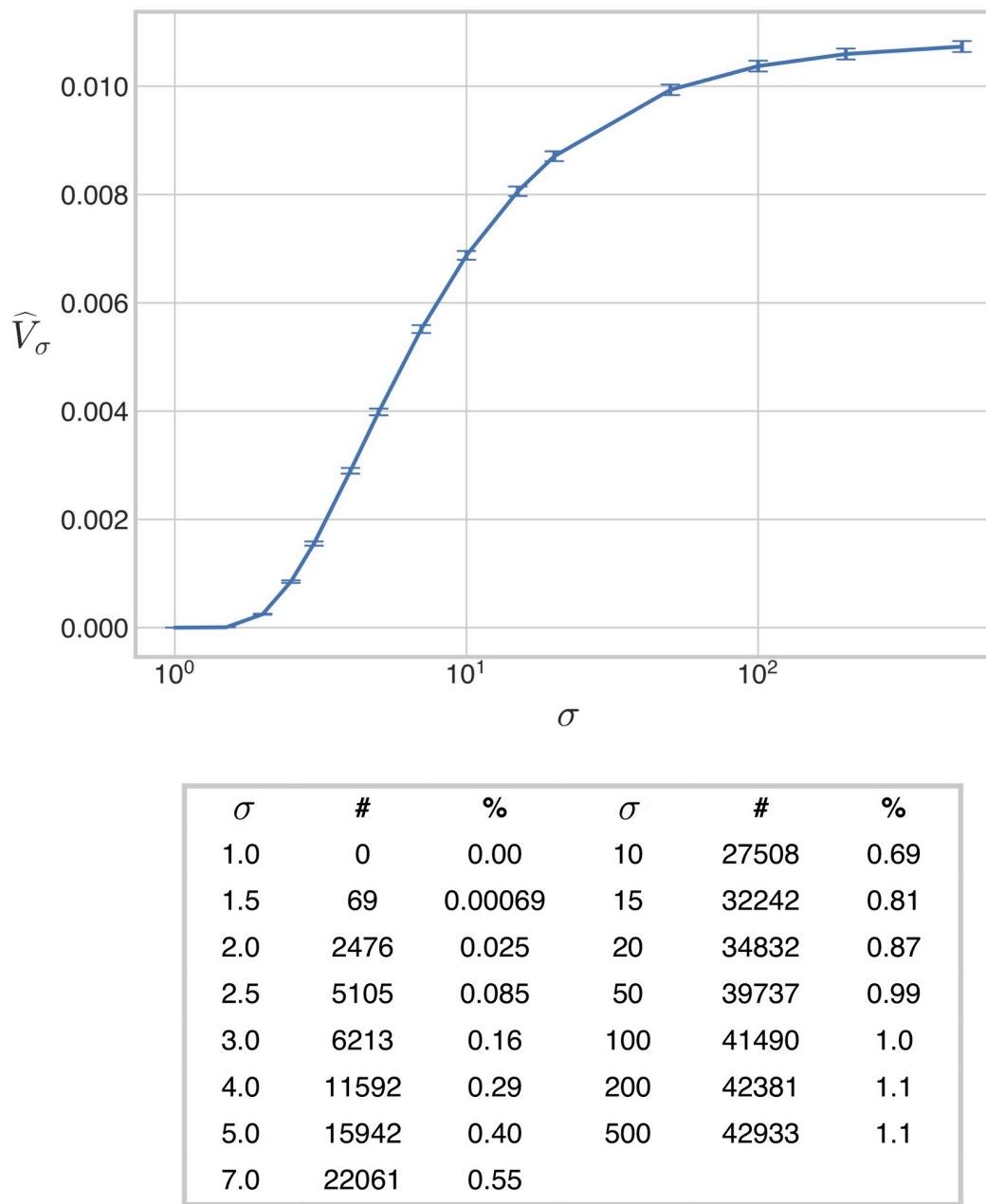

**Fig 3. Volume of the bistable region under weak irreversibility.** The 8-dimensional volume of the bistable region, normalised as a proportion of the volume of the box $\mathcal{H}$, is plotted against the values of $\sigma$ in Eq 11, for the case when the reactions $S_1 \xrightarrow{E} S_2$ and $S_1 \xrightarrow{F} S_0$ are weakly irreversible. The accompanying table lists the number of bistable points found for each value of $\sigma$, together with the percentage of the box $\mathcal{H}$ occupied by the bistable region. The error bars give 95% confidence intervals for each estimate (Materials and methods). The estimates have been joined by line segments.

The results of the estimation are shown in Fig 3. We first found that the estimated normalised volume of the bistable region is zero at $\sigma = 1.0$. This suggests the existence of a threshold in $\sigma$, below which there is no bistability; we explore this possibility further below. The normalised volume then appears to increase smoothly in a sigmoidal ("S-shaped") manner and

saturate at large values of $\sigma$. Saturation was not expected on mathematical grounds (Discussion) and we found that it is more apparent under assumptions of weak irreversibility than under strong irreversibility (below). The value of $\sigma$ at which saturation is established, and the saturating volume itself, are difficult to determine precisely, but our analysis suggests that the saturating volume is close to 1.1% of the volume of $\mathcal{H}$. We were surprised by how small this was. It suggests that, under realistic assumptions of weak irreversibility, bistability is robust but rare. We return to this point in the Discussion.

## A threshold for bistability

The curve in Fig 3 raises several questions. First, it is unclear whether the lack of any observed bistability at $\sigma = 1.0$ reflects the existence of a <u>bona fide</u> threshold for bistability, $\sigma^*$, below which there is only monostability, or rather arises as a consequence of undersampling. To clarify this point, we sought further evidence that $\mathcal{M}_1$ is indeed empty. We reasoned that, if bistability does exist for values of $\sigma$ near 1, it is more likely to occur near those parameter points at which bistability occurs for larger values of $\sigma$ (see also the section below on "blinking"). To find such points systematically, we turned to importance sampling, as implemented in the VEGAS algorithm, introduced originally for Monte Carlo estimation of multi-dimensional integrals [84].

VEGAS starts from an initial sample of points in a subregion of a multi-dimensional space and adaptively constructs augmented samples that are preferentially drawn from regions of higher sample density. Specifically, the section of each coordinate axis containing the projection of the sample is partitioned into a specified number, $M$, of bins whose lengths are chosen so that each bin contains the same number of projected sample points, up to some smoothing of the bin frequencies (Materials and methods). There are, therefore, proportionately more smaller bins in regions of higher sample density. Subsequently, a new sample of $N$ points is generated one coordinate at a time, with each coordinate of each point chosen uniformly at random from a bin along the corresponding axis, such that the $N$ values are roughly evenly partitioned among the $M$ bins. This results in a new sample of $N$ points biased towards high-density regions of the initial sample. The initial sample is then augmented with the new sample, and the entire process repeated $T$ times.

We first generated 12 VEGAS samples, one for each of the following values of $\sigma$,

$$\sigma = 2.5 \, , \; 3.0 \, , \; 4.0 \, , \; 5.0 \, , \; 7.0 \, , \; 10 \, , \; 15 \, , \; 20 \, , \; 50 \, , \; 100 \, , \; 200 \, , \; 500 \, . \tag{14}$$

For each value of $\sigma$ listed in Eq 14, we initialised the VEGAS algorithm using the set of bistable points gathered through ILR sampling at the given value of $\sigma$. For instance, the set of 27508 bistable points found through ILR sampling at $\sigma = 10$ (Fig 3) was used as an initial sample for a VEGAS sample at $\sigma = 10$. We generated each VEGAS sample over $T = 6$ iterations, during each of which we sampled $N = 10^6$ parameter points and determined the subset of bistable points at the given value of $\sigma$ using Bertini and Paramotopy. All sampling was performed over logarithmic coordinates, with each coordinate sampled from $M = 50$ bins covering the interval $[-1, 1]$. Thus, we obtained 12 VEGAS samples, one for each value of $\sigma$ listed in Eq 14, each containing $6 \times 10^6$ points.

As expected for importance sampling, the proportion of bistable points found within each VEGAS sample at the corresponding value of $\sigma$ was much larger than $\widehat{V}_\sigma$. The number of bistable points in each VEGAS sample ranged between $\sim 2.45 \times 10^6$ (at $\sigma = 2.5$) and $\sim 3.35 \times 10^6$ (at $\sigma = 500$), out of a possible $6 \times 10^6$. In marked contrast, running Bertini and Paramotopy on all 12 VEGAS samples at $\sigma = 1.0$ found zero bistable points, out of $7.2 \times 10^7$ points. This further suggests that $\mathcal{M}_1$ is indeed empty.

As a further test for this hypothesis, we generated one additional VEGAS sample for smaller values of $\sigma$. Here, we reasoned that the few bistable points found through ILR sampling for $\sigma = 1.5$ and $\sigma = 2.0$ (69 and 2476, respectively) rendered these sets inadequate for initialising the VEGAS algorithm. Thus, we instead opted to use the set of $\sim 2.45 \times 10^6$ bistable points found in the VEGAS sample for $\sigma = 2.5$ as an initial sample, and generated a VEGAS sample of $M = 6 \times 10^6$ parameter points in a single ($T = 1$) iteration, again sampling each parameter from $M = 50$ bins covering the interval $[-1, 1]$ in logarithmic coordinates. This VEGAS sample contained numerous bistable points at $\sigma = 1.5$ and $\sigma = 2.0$ ($\sim 1.79 \times 10^5$ and $\sim 1.48 \times 10^6$, respectively), but did not contain any bistable points at $\sigma = 1.0$.

In sum, the above results suggest that $\mathcal{M}_1$ is indeed empty, and that a threshold for bistability, $1 \leq \sigma^* < 1.5$, within $\mathcal{H}$ does exist. We believe this is an instance of a more general mathematical result (Discussion). We note that this conclusion is limited by the choice of box, $\mathcal{H}$, we used to bound the parameter values. Indeed, it is entirely possible that there exist parameter points outside $\mathcal{H}$ that exhibit bistability at $\sigma = 1$, or even at arbitrarily small values of $\sigma$. We address this possibility in a subsequent section.

## Bistable regions are connected

As explained in the Introduction, the positive volume of the bistable region confirms the simplest requirement for robustness but does not tell us about the shape of the region (Fig 1B). We therefore sought evidence for whether the bistable region $\mathcal{M}_\sigma$ is topologically connected, at least for $\sigma \geq 1.5$ for which we know it is non-empty (Fig 3). The challenge here is that we have to assess connectedness from finite samples of parameter points, $\mathcal{S} \subset \mathcal{H}$, which provide only discrete approximations, $\widehat{\mathcal{M}}_\sigma = \mathcal{S} \cap \mathcal{M}_\sigma$, of the bistable region.

To address this, we constructed the <u>connectivity graph</u>, $\mathcal{G}_\Delta(\widehat{\mathcal{M}}_\sigma)$, associated with the sample. We adapted this idea from previous studies of robotic motion planning, in which the graph is used to determine connected, obstacle-free regions of a robot's configuration space [85, 86]. The vertices of the connectivity graph are the bistable points, $\widehat{\mathcal{M}}_\sigma$, in the sample, and there is an undirected edge between two vertices if they are within Euclidean distance $\Delta$ of each other in logarithmic coordinates. Here, $\Delta > 0$ is an adjustable threshold. Such a graph can be partitioned into <u>connected components</u>. Any two vertices within the same connected component can be joined by a path of contiguous edges, while no such path exists between vertices in different connected components.

Consider a sufficiently large finite sample of some connected region in a multi-dimensional space. If $\Delta$ is larger than the maximum distance between two points, then every point is connected to every other point and the graph consists of a single connected component. If $\Delta$ is smaller than the minimum distance between two points, then no point is connected to any other and there are as many connected components as there are vertices. In between these extremes, however, we would expect a different behaviour, with a single very large connected component, comprising most of the points in the interior of the region, together with many much smaller components, typically comprising points close to the boundary of the region that fail to be within $\Delta$ of the largest component.

In contrast, consider a region that is disconnected and consists of several connected components. We would then expect that for intermediate values of $\Delta$, the corresponding connectivity graph would break up into several large connected components, along with many much smaller boundary components. Unless some of the connected components of the region were much smaller than others, we would expect each such component to manifest as a large connected component in the graph. We can thereby estimate the number of the former by counting instances of the latter.

It is not straightforward to assess the statistical accuracy of such estimates. To do so requires specifying a prior expectation for the region's connectivity, and we have little to guide us in knowing what to expect of parameter geography. It is easy to imagine complicated regions, such as those with one very large part and many very small ones, that would confuse a connectivity graph analysis. Also, sampling is oblivious to regions of lower dimension, as noted in the Introduction, and we build the graphs from points within the finite box $\mathcal{H}$, both of which issues could compromise the conclusions drawn from a connectivity graph analysis. With these caveats in mind, we believe the structure of the connectivity graph for intermediate values of $\Delta$ provides helpful preliminary evidence for the connectivity of the bistable region. There are also further tests that we can undertake, as explained below.

The connectivity graph is most informative when there are many sample points, so we used the bistable sets obtained from the VEGAS samples described above to build connectivity graphs, $\mathcal{G}_\Delta(\widehat{\mathcal{M}}_\sigma)$, for each of the values

$$\sigma = 1.5,\ 2.0,\ 2.5,\ 3.0,\ 4.0,\ 5.0,\ 7.0,\ 10,\ 15,\ 20,\ 50,\ 100,\ 200,\ 500. \tag{15}$$

The graphs for $\sigma = 1.5$ and $\sigma = 2.0$ were built from the single-iteration VEGAS sample gathered for small $\sigma$; for each remaining value of $\sigma$, the graph was built from the corresponding six-iteration VEGAS sample.

Constructing the graph in its entirety is computationally intractable if the sample size is large because the number of edges scales quadratically with the number of vertices. However, it is sufficient to construct a spanning forest, an acyclic subgraph that includes every vertex of the full graph. This is because the connected components of the spanning forest are individual spanning trees that are in bijective correspondence with the connected components of the full graph. A full description of this algorithm is given in the Materials and Methods.

Given a set of bistable points gathered from an ILR sample, $\mathcal{S} \subset \mathcal{H}$, it is straightforward to estimate a value of $\Delta$ for which, given a point in $\mathcal{S}$, there is a large probability, say 0.99, that there is at least one other point in $\mathcal{S}$ within distance $\Delta$. This seems a reasonable choice for an intermediate value of $\Delta$ because it allows most vertices in the graph to have an incident edge. It is shown in the Materials and Methods how to calculate such a $\Delta$ for ILR samples of a given size. The VEGAS samples were not constructed by ILR sampling but each augmented VEGAS sample $\mathcal{S}_A$ is generated from an initial set of bistable points obtained from an ILR sample, $\mathcal{S}_I$. We therefore chose an effective sample size, $N'$, that scaled proportionally with the increase in the number of bistable points,

$$N' = \left( \frac{\#\text{bistable points in } \mathcal{S}_A}{\#\text{bistable points in } \mathcal{S}_I} \right) \#\mathcal{S}_I,$$

and used $N'$ in place of $\#\mathcal{S}$ as the sample size in calculating $\Delta$. We found using this approach that $\Delta = 0.15$ is a suitable choice for all values of $\sigma$ given in Eq 15 (Materials and methods).

The results of the connectivity graph analysis are shown in Table 1. For each value of $\sigma$ in Eq 15, the largest connected component of the graph contains the vast majority of vertices. For instance, for $\sigma = 3.0$, there are 2705883 bistable points in the sample, which decompose into 122672 connected components. Of these, the largest component contains 2556447 vertices, or $\sim$94% of the total, while the second largest component contains a mere 21 vertices. While the proportion of vertices in the largest component decreases with $\sigma$—at $\sigma = 500$, only $\sim$74% of the vertices lie in the largest component—the size of the second largest connected component remains tiny, relative to that of the largest component, as $\sigma$ increases. Furthermore, the number of components rapidly increases with $\sigma$, and a roughly constant percentage of them, between $\sim$83% and $\sim$87%, are singletons, indicating that the connectivity graph decomposes

**Table 1. Connectivity of the bistable region.** Details of the initial connectivity graphs for the indicated values of $\sigma$ in column 1. Column 2 gives the number of bistable points in the set used to construct the graph; column 3 gives the number of connected components in the resulting connectivity graph; columns 4 and 5 give the sizes of the largest and second-largest components of the graph, respectively; column 6 gives the number of singleton components; and column 7 gives the size of the largest component as a percentage of the number of bistable points in the sample. The numbers show that the connectivity graphs consist of one large component and many very small components, suggesting that the bistable region is connected for all examined values of $\sigma$. Details of the connectivity graphs obtained through refinement with additional sampling (Results, Materials and methods) are given in S1 Appendix.

| $\sigma$ | $\widehat{\mathcal{M}}_\sigma$ | # components | #$C_1$ | #$C_2$ | # singletons | $C_{1\#}/\widehat{\mathcal{M}}_\sigma$ |
|---|---|---|---|---|---|---|
| 1.5 | 179130 | 4519 | 173773 | 12 | 3923 | 97% |
| 2.0 | 1481421 | 29484 | 1446437 | 16 | 25788 | 98% |
| 2.5 | 2456156 | 76896 | 2362892 | 22 | 66180 | 96% |
| 3.0 | 2705883 | 122672 | 2556447 | 21 | 105260 | 94% |
| 4.0 | 2932035 | 209165 | 2672497 | 36 | 177833 | 91% |
| 5.0 | 3050788 | 278684 | 2702553 | 28 | 235930 | 89% |
| 7.0 | 3146912 | 373650 | 2672992 | 35 | 314273 | 85% |
| 10 | 3236189 | 452663 | 2659036 | 33 | 379762 | 82% |
| 15 | 3271113 | 527437 | 2592576 | 38 | 440444 | 79% |
| 20 | 3303594 | 563899 | 2576185 | 55 | 470983 | 78% |
| 50 | 3366306 | 631931 | 2548744 | 54 | 527579 | 76% |
| 100 | 3382039 | 658003 | 2529828 | 42 | 548689 | 75% |
| 200 | 3401104 | 669140 | 2533572 | 62 | 558110 | 74% |
| 500 | 3396987 | 677710 | 2519055 | 38 | 565368 | 74% |

into one huge component and increasing numbers of very tiny components. This is exactly what would be expected if $\mathcal{M}_\sigma$ were connected.

If this is in fact the case, we should be able to connect the vertices in the smaller connected components to the largest component by paths of bistable points. Accordingly, we undertook a further test of the initial connectivity graphs described above (Table 1). For each graph, we considered each of the non-largest components and chose a vertex, $\theta$, uniformly at random in that component. We then selected the $K$ (approximate) nearest neighbours, $v^{(1)}, \ldots, v^{(K)}$, to $\theta$ from the largest component, and sampled along the straight line segments between $\theta$ and $v^{(j)}$, for $j = 1, \ldots, K$, by sub-dividing each line segment into sub-intervals of length $0.98\Delta$ and collecting the endpoints of these sub-intervals. We used Bertini and Paramotopy to determine which of these sampled points were in the bistable region. We then recomputed the connectivity graph with these newfound bistable points added to the original bistable set, and iterated this procedure until the proportion of vertices in the largest component remained constant between consecutive iterations.

The computationally intensive step here is determining which point from the largest component is nearest to each randomly selected vertex, $\theta$. For this, we used the Approximate Nearest Neighbour (ANN) algorithm [87], which builds a hierarchical data structure to efficiently select $K$ points, $v^{(1)}, \ldots, v^{(K)}$, which are approximately nearest to $\theta$ in the sense that

$$d(\theta, v^{(j)}) \leq (1 + \epsilon)\, d(\theta, \mu^{(j)}) \qquad \text{for } j = 1, \ldots, K, \tag{16}$$

where $d$ is the distance metric being used (in our case, Euclidean distance over logarithmic coordinates) and $\mu^{(j)}$ is the true $j$th nearest neighbour to $\theta$, for $j = 1, \ldots, K$. The approximation factor, $\epsilon$, can be chosen to be as small as required and the algorithm is optimal in an appropriate sense [87]. Experience with the open-source C++ ANN library suggests that this is an efficient procedure for samples of order $10^5$ in spaces of dimension up to 20, and that the probability of choosing a non-nearest neighbour is relatively small in practice [88].

For our analysis, we used an approximation factor of $\epsilon = 0.001$, increasing $K$ after each iteration (Materials and methods). Within two iterations of this procedure, we were able to obtain, for each of the values of $\sigma$ in Eq 15, a set of bistable points whose connectivity graph consisted of a single connected component. This provides yet further evidence that the bistable region is connected. The full results of this analysis are given in S1 Appendix, Table C.

## Bistable regions are not convex but have high visibility ratios

As described in the Introduction, convexity tells us about the shape of a region and rules out many features that compromise robustness, like the waists and holes in Fig 1B. It is not difficult to show, by randomly sampling pairs of points within the bistable regions, that none of these regions are convex. But how far do they depart from convexity? The <u>visibility ratio</u> of a region offers a measure of this. Informally, it is the probability that the line joining two points drawn at random from the region lies entirely in the region. Formally, we define the visibility ratio of $\mathcal{M}_\sigma$ as

$$\mathrm{vis}(\mathcal{M}_\sigma) = \frac{1}{V_\sigma^2} \int_{(\theta,\mu)\in\mathcal{M}_\sigma\times\mathcal{M}_\sigma} v(\theta,\mu) \; d\left(\log\,\theta,\log\,\mu\right), \qquad (17)$$

where $v : \mathcal{M}_\sigma \times \mathcal{M}_\sigma \to \{0, 1\}$ is the indicator function of the set of pairs of bistable points for which the straight line segment (over logarithmic coordinates) between the two points lies entirely within the bistable region,

$$v(\theta,\mu) = \begin{cases} 1 & \text{if } \iota_\sigma(10^{t\,\log\,\theta+(1-t)\,\log\,\mu}) = 1 \text{ for } 0 \le t \le 1 \\ 0 & \text{otherwise.} \end{cases}$$

With the normalisation in Eq 17, it is easy to see that, if $\mathcal{M}_\sigma$ is convex, then $\mathrm{vis}(\mathcal{M}_\sigma) = 1$.

Since computing $v(\theta, \mu)$ for even a single choice of $\theta$ and $\mu$ requires evaluating $\iota_\sigma$ on infinitely many parameter points, we cannot directly estimate $\mathrm{vis}(\mathcal{M}_\sigma)$. Therefore, we sought to approximate $v(\theta, \mu)$ by sub-dividing the straight line segment between $\theta$ and $\mu$ into $K + 1$ sub-intervals of equal length (over logarithmic coordinates), and checking bistability only on the $K$ endpoints of the intervals. In other words, we computed the following indicator function,

$$v_K(\theta,\mu) = \begin{cases} 1 & \text{if } \iota_\sigma(10^{t\,\log\,\theta+(1-t)\,\log\,\mu}) = 1 \text{ for all } t = \frac{j\,d(\theta,\mu)}{K+1}, j = 1,\dots,K \\ 0 & \text{otherwise.} \end{cases}$$

This quantity converges to $v(\theta, \mu)$ as $K \to \infty$. Integrating over $\mathcal{M}_\sigma \times \mathcal{M}_\sigma$ and normalising, we obtain the <u>$K$-fold visibility ratio</u>,

$$\mathrm{vis}(\mathcal{M}_\sigma, K) = \frac{1}{V_\sigma^2} \int_{(\theta,\mu)\in\mathcal{M}_\sigma\times\mathcal{M}_\sigma} v_K(\theta,\mu) \; d(\log\,\theta,\log\,\mu), \qquad (18)$$

which converges to $\mathrm{vis}(\mathcal{M}_\sigma)$ as $K \to \infty$.

Since $v_K(\theta, \mu) = 0$ implies that $v(\theta, \mu) = 0$, $\mathrm{vis}(\mathcal{M}_\sigma, K) \ge \mathrm{vis}(\mathcal{M}_\sigma)$ for all $K > 0$. The $K$-fold visibility ratio will therefore tend to overestimate the true visibility ratio. With this caveat in mind, it offers a computationally feasible estimate for the extent of departure from convexity.

We estimated the $K$-fold visibility ratio by randomly choosing $M$ pairs of bistable points

$$\left(\theta^{(1)},\mu^{(1)}\right), \; \dots, \; \left(\theta^{(M)},\mu^{(M)}\right) \in \widehat{\mathcal{M}}_\sigma \times \widehat{\mathcal{M}}_\sigma$$

without replacement, where $\widehat{\mathcal{M}}_\sigma$ is a set of bistable points gathered with ILR sampling from $\mathcal{H}$,

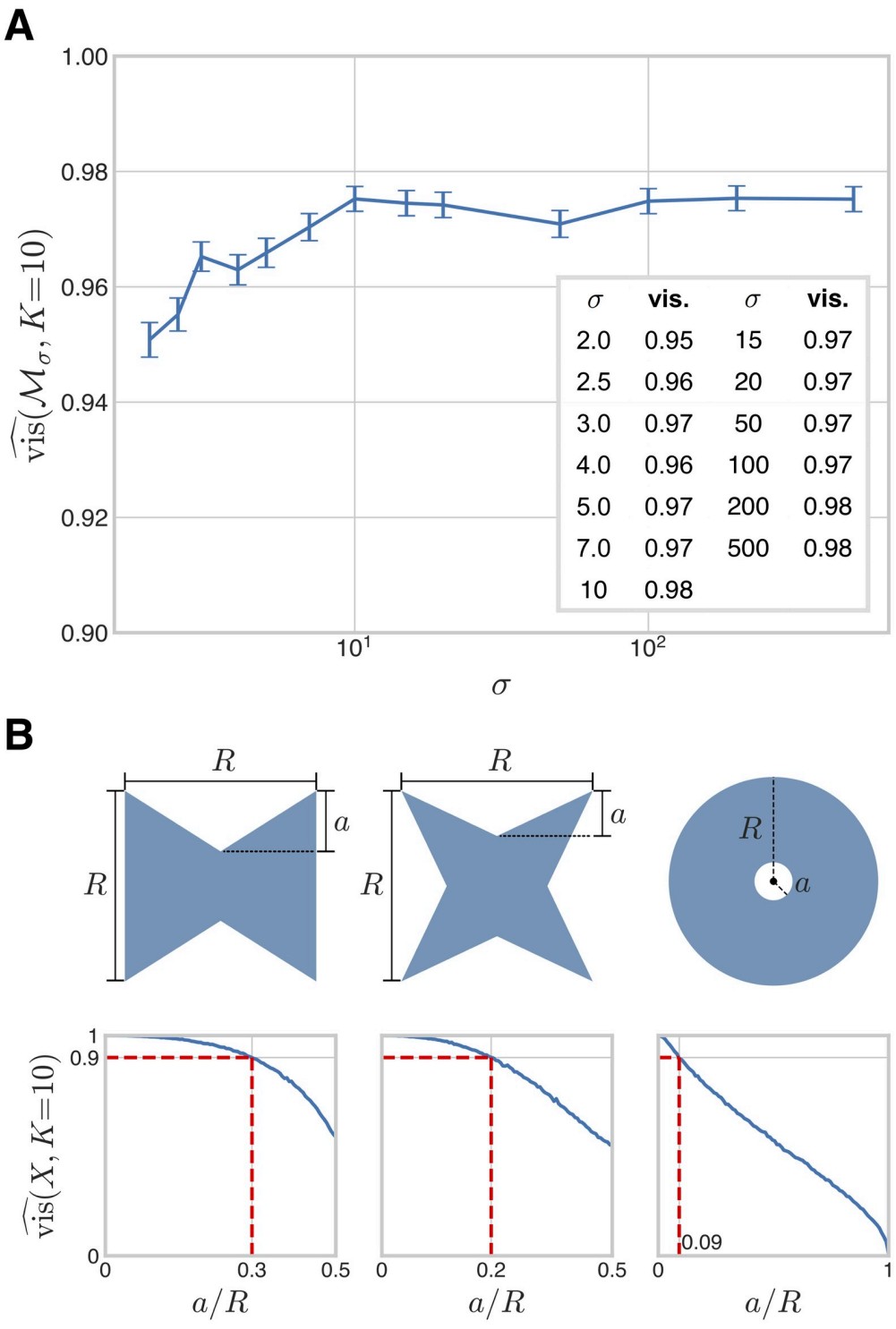

**Fig 4. Visibility ratio of the bistable region under weak irreversibility.** A: The 10-fold visibility ratio of the bistable region under weak irreversibility is plotted as a function of $\sigma$. B: Three families of 2-dimensional regions, with length parameters $a$ and $R$, as shown, along with plots of their 10-fold visibility ratios as functions of $a/R$, with $R = 1$ and $a$ varying. 10-fold visibility ratios were numerically computed using $M = 20000$ random pairs of points, as in the main text. The dotted red lines in the plots indicate the values of $a/R$ at which the visibility ratio is 0.9.

and computing the estimator

$$\widehat{\text{vis}}(\mathcal{M}_\sigma, K) = \frac{1}{M} \sum_{j=1}^{M} v_K(\theta^{(j)}, \mu^{(j)}).$$

We undertook this estimation by choosing $K = 10$ and sampling $M = 20000$ pairs of points, without replacement, from the bistable points gathered through ILR sampling for each of the values of $\sigma$ given in Eq 11, except for $\sigma = 1.0$ and $\sigma = 1.5$, at which too few bistable points were found through ILR sampling (Fig 3; see also the previous section on the bistable volume). Confidence intervals for these estimates were computed using standard results in finite population sampling statistics (Materials and methods).

   The results of this analysis are shown in Fig 4A. We found that although the bistable region is not convex for any value of $\sigma$, it is close to convex for all considered values of $\sigma$, with 10-fold visibility ratios exceeding 0.95. This suggests that, for a large majority of pairs of bistable parameter points, the straight line connecting them in parameter space consists also of bistable points.

   The visibility ratio is a global measure of the region's departure from convexity but offers little information regarding what local geometric features may influence the loss of convexity. Different shapes can have the same visibility ratio and how the visibility ratio depends on shape is difficult to describe in general. Low-dimensional examples suggest that interior holes reduce the visibility ratio to a greater extent than boundary indentations (Fig 4B). The fact that the visibility ratio appears to be largely independent of $\sigma$ suggests two possibilities. Either the bistable region grows radially outward in parameter space, in such a way that the size of any local geometric feature which compromises convexity scales with that of the entire bistable region; or these local geometric features appear and disappear in a manner that approximately preserves the visibility ratio as $\sigma$ increases. The latter possibility may appear less plausible but the next section suggests that the growth of the bistable region with $\sigma$ may indeed exhibit phenomena of this kind.

## Non-monotonic growth of the bistable region

The volume of the bistable region, $V_\sigma$, appears to increase monotonically as $\sigma$ increases (Fig 3). This monotonicity would follow naturally if, once a parameter point $\theta \in \mathcal{H}$ exhibits bistability at some value $\sigma = a$, it continues to do so for all $\sigma > a$. The bistable region would then increase in extent with $\sigma$, so that if $a < b$, then $\mathcal{M}_a \subset \mathcal{M}_b$. We therefore tested this behaviour and were surprised to find, in contrast to our initial expectation, that parameter points do not behave this way. Instead, they can transition back and forth between monostability and bistability.

   To illustrate the possibilities, Fig 5 shows four parameter points which exhibit different behaviours as $\sigma$ takes the values 3.0, 4.0, 5.0, 7.0, and 10. Each plot shows the "pseudo-null-clines," $\Phi_1(u, v) = 0$ and $\Phi_2(u, v) = 0$ from Eq 10, for which the intersections of the two curves give the steady-states of the PTM system [30]. (The zero solution, $u = v = 0$, is isolated from these curves in the real $uv$-plane and is omitted for simplicity.) The first parameter point,

$$\begin{aligned} \theta^{(1)} \quad &= (7.239380624,\ 0.1047069812,\ 0.1045129736,\ 1.114909161, \\ &\quad 2.017259480,\ 0.1057288524,\ 2.544613812,\ 0.1021803160), \end{aligned}$$

is monostable at all five values of $\sigma$. The second parameter point,

$$\begin{aligned} \theta^{(2)} \quad &= (4.684802750,\ 0.5200773771,\ 1.589343505,\ 0.1804775460, \\ &\quad 0.1270218587,\ 0.1129848293,\ 0.1816907000,\ 5.708851611), \end{aligned}$$

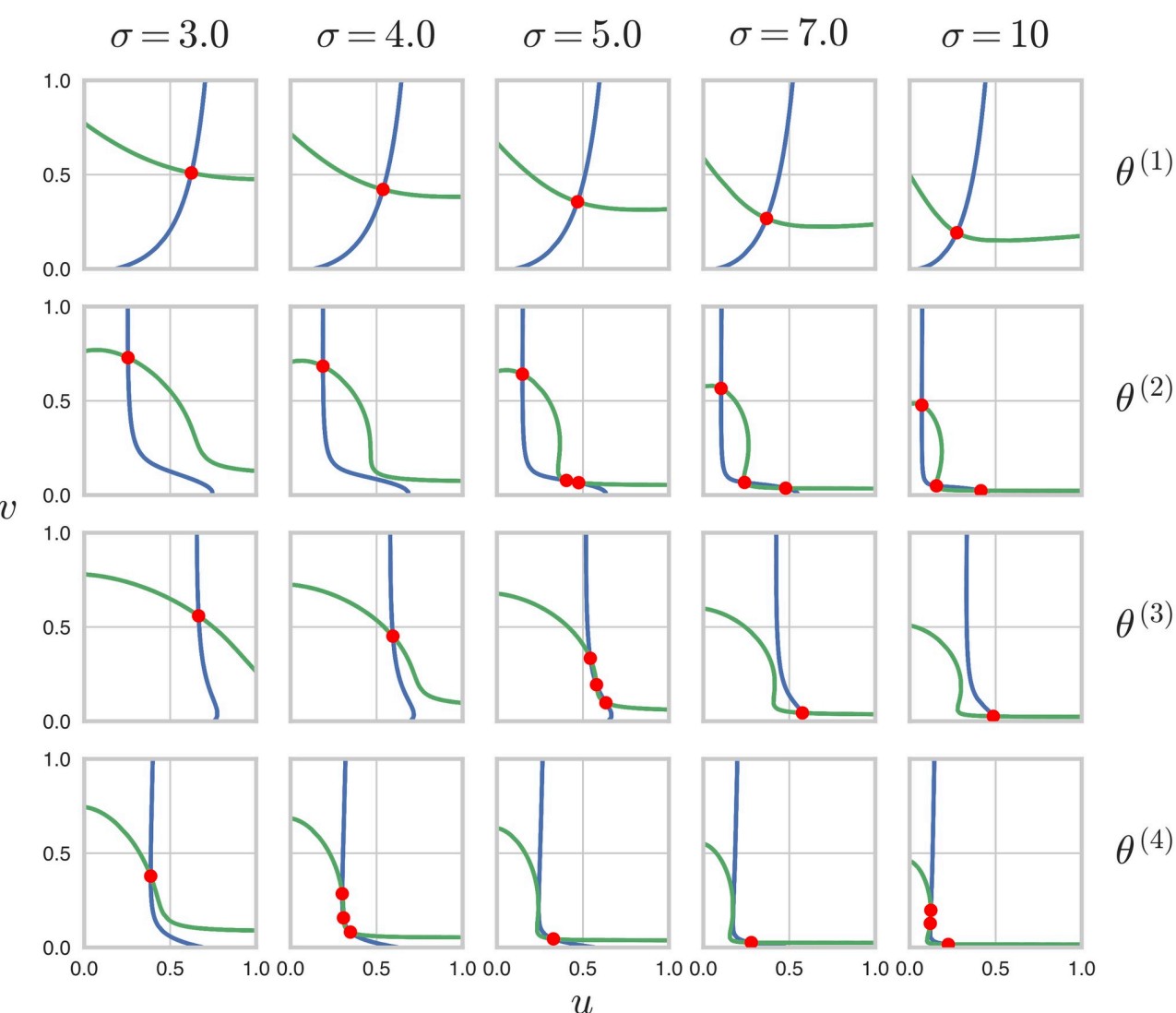

**Fig 5. Parameter points move between monostability and bistability.** Pseudo-nullcline plots for the steady-state equations, $\Phi_1(u, v) = 0$ (blue) and $\Phi_2(u, v) = 0$ (green), in Eq 10 are shown for the four parameter values given in the main text, at five values of $\sigma$, showing how steady-states (red dots) appear and disappear at the intersections of the pseudo-nullclines. The zero solution, $u = v = 0$ is isolated from the pseudo-nullclines in the real $uv$-plane and is not shown for simplicity.

becomes bistable between $\sigma = 4.0$ and $\sigma = 5.0$, and remains so at $\sigma = 5.0$, $\sigma = 7.0$, and $\sigma = 10$. The third parameter point,

$$\theta^{(3)} = (6.865206499, 0.2105440222, 0.2524064999, 0.1025041753,$$
$$0.1178107732, 0.1017610135, 0.4495453146, 5.796119139),$$

becomes bistable between $\sigma = 4.0$ and $\sigma = 5.0$, then reverts to monostability between $\sigma = 5.0$ and $\sigma = 7.0$. Finally, the fourth parameter point,

$$\theta^{(4)} = (4.885859541, 0.4985163876, 0.7212641184, 2.119877596,$$
$$0.1655484166, 0.1246565927, 0.7180476375, 8.912153306),$$

**Table 2. Blinking parameter points.** For each value of $\sigma$ in column 1, the number of bistable points in the $4 \times 10^6$ samples obtained by independent logarithmic random (ILR) sampling is shown in column 2 (see also Fig 3). Of these bistable points, the number of blinking points, or those which become monostable at some larger value of $\sigma$, is shown in column 3 (titled BP). The subsequent columns indicate, for each value of $\sigma$, the number of blinking points that lie outside the largest component in the corresponding initial connectivity graph (BPSC, column 4); the number of blinking points that never enter the largest component of the initial connectivity graph for larger values of $\sigma$ (BPNL, column 5); and the number of blinking points that are monostable at $\sigma = 500$ ("asymptotically monostable", BPAM, column 6). The numbers show that blinking points are found primarily on the boundary of the bistable region and become monostable for large values of $\sigma$, as discussed further in the text.

| $\sigma$ | $\#\widehat{\mathcal{M}}_\sigma$ | # BP | # BPSC | # BPNL | # BPAM |
|---|---|---|---|---|---|
| 1.5 | 23 | 0 | 0 | 0 | 0 |
| 2.0 | 992 | 8 | 8 | 3 | 2 |
| 2.5 | 3451 | 62 | 62 | 40 | 36 |
| 3.0 | 6213 | 94 | 94 | 67 | 70 |
| 4.0 | 11592 | 230 | 230 | 180 | 205 |
| 5.0 | 15942 | 308 | 308 | 260 | 290 |
| 7.0 | 22061 | 381 | 381 | 313 | 370 |
| 10 | 27508 | 396 | 396 | 348 | 392 |
| 15 | 32242 | 337 | 337 | 288 | 334 |
| 20 | 34832 | 299 | 299 | 281 | 298 |
| 50 | 39737 | 169 | 169 | 157 | 169 |
| 100 | 41490 | 82 | 82 | 76 | 82 |
| 200 | 42381 | 34 | 34 | 34 | 34 |

becomes bistable between $\sigma = 3.0$ and $\sigma = 4.0$, reverts to monostability between $\sigma = 4.0$ and $\sigma = 5.0$, then reverts back to bistability between $\sigma = 7.0$ and $\sigma = 10$.

We examined the common ILR sample of $4 \times 10^6$ points in $\mathcal{H}$ at which solutions to Eq 10 were obtained for each value of $\sigma$ in Eq 11. We determined for each value of $\sigma$ the subset of bistable parameter points which become monostable at some larger value of $\sigma$ (Table 2). We refer to this phenomenon as "blinking." We were able to alphaCertify the solutions to Eq 10 at each of these parameter points for all values of $\sigma$, thereby confirming that the designations of monostability or bistability were mathematically correct. Blinking is therefore not a numerical artifact (Materials and methods).

If blinking points occur within the interior of the bistable region, it suggests that the region can develop interior holes. This would be surprising in view of the high visibility ratio found previously, as such features tend to lower the visibility ratio (Fig 4B). We therefore examined the blinking points in relation to the connectivity graph $\mathcal{G}_\Delta(\mathcal{M}_\sigma)$ of the bistable region (Table 1, S1 Appendix) and summarised the findings in Table 2. The number of blinking parameter points increases from 8 at $\sigma = 2$ to a maximum of 396 at $\sigma = 10$ but then decreases to 34 at $\sigma = 200$ (Table 2, column 3), despite the monotonic increase in the volume of the bistable region (Fig 3). Every blinking point was found to lie outside the largest connected component of the corresponding connectivity graph for every value of $\sigma$ (Table 2, column 4). For any given value of $\sigma$, a sizeable majority of blinking points never enter the largest connected component of the graph at larger values of $\sigma$ (Table 2, column 5). For instance, among the 308 blinking points at $\sigma = 5$, 260 (84%) never enter the largest component. These findings suggest that bistable points that become monostable at larger values of $\sigma$ lie on the boundary of the bistable region, rather than in its interior, and tend to remain near the boundary as they exit and enter the bistable region as $\sigma$ increases. Finally, most blinking points become monostable at $\sigma = 500$ (Table 2, column 6). For instance, of the 308 blinking points at $\sigma = 5$, 290 (94%) are

monostable at $\sigma = 500$. This may indicate that many blinking points become asymptotically monostable as $\sigma \to \infty$. We return to these interesting findings in the Discussion.

## The strongly irreversible case

Up to now, we have assumed that the enzymes in the PTM system are irreversible and that the reactions in which $S_1$ is a substrate, $S_1 \xrightarrow{E} S_2$ and $S_1 \xrightarrow{F} S_0$, are also weakly irreversible, with non-zero affinity for product rebinding. (Recall that the reactions in which $S_1$ is a product, $S_0 \xrightarrow{E} S_1$ and $S_2 \xrightarrow{F} S_1$, may be either strongly or weakly irreversible without affecting the signs of the non-dimensional parameters (Eq 9) and the results deduced so far.) As explained in the Introduction, weak irreversibility is the realistic assumption for enzymes in a PTM system. However, in view of the nearly universal reliance in the literature on the Michaelis-Menten reaction scheme, we wanted to understand the impact of strong irreversibility on parameter geography.

Accordingly, we assume now that the reactions in which $S_1$ is a substrate are strongly irreversible, so that $\epsilon_2 = \phi_0 = 0$, and take the parameter point $\eta \in \mathbb{R}^6$ to be defined by

$$\eta_1 = \alpha \; , \;\; \eta_2 = \beta \; , \;\; \eta_3 = \epsilon_0 \; , \;\; \eta_4 = \epsilon_1 \; , \;\; \eta_5 = \phi_1 \; , \;\; \eta_6 = \phi_2 \, .$$

We consider solutions within the finite-volume box $\mathcal{H}^* = [0.1, 10]^6$. The steady-state polynomial equations in Eq 10 still have degree 4 and Bézout's Theorem tells us that there are 16 complex solutions in projective space. Given $\zeta = 1$, a fixed value of $\sigma$ and a randomly chosen parameter point in $\mathcal{H}^*$, the generic solutions are as follows: the zero solution, $u = v = 0$, which has multiplicity 6; five additional finite solutions; and five solutions which are projectively at infinity. Of the five nonzero, finite solutions, we always find either one or three positive real solutions, which we refer to as monostability and bistability, respectively, as explained previously. Let $\mathcal{M}_\sigma^* \subset \mathcal{H}^*$ denote the subset of bistable parameter points at a given value of $\sigma$.

The volume, $V_\sigma^*$, of $\mathcal{M}_\sigma^*$ and a statistical estimator of this volume, $\widehat{V}_\sigma^*$, can be defined in a similar way to Eqs 12 and 13. Here, we used a single sample of $4 \times 10^6$ parameter points, generated by ILR sampling from $\mathcal{H}^*$, to compute $\widehat{V}_\sigma^*$, as we found sufficiently many bistable points at all values of $\sigma$ to ensure high statistical confidence. The results are shown in Fig 6. As in the weakly irreversible case (Fig 3), the volume appears to increase monotonically with $\sigma$. The threshold for bistability appears, as before, to be at or slightly above $\sigma = 1$. There are, however, two important differences between the weakly and strongly irreversible cases. First, saturation is less apparent under strong irreversibility, even with additional volumes evaluated at $\sigma = 1000, 2000$, and $5000$. Second, under strong irreversibility the bistable volume increases much more rapidly with increasing $\sigma$: at $\sigma = 500$, the bistable region occupies 20% of $\mathcal{H}^*$ under strong irreversibility in contrast to only 1.1% of $\mathcal{H}$ under weak irreversibility.

Volumes scale in different ways with the dimension of the ambient space. The volume of a ball of radius $r$ goes to zero as $n \to \infty$. In contrast, the volume of a hypercube of length $r$ scales as $r^n$. Because we have used the volume of a hypercube with sides $[0.1, 10]$ to normalise the volume of the bistable region, we expect this to compensate for the change in the ambient space from six dimensions to eight, even if we do not know how to accommodate the shape of the bistable region in the normalisation. With this caveat, the decrease in the normalised volume from strong to weak irreversibility still seems dramatic. Bistability appears to be a far more robust property under the unrealistic assumption of strong irreversibility, such as with Michaelis-Menten enzyme mechanisms, than it does under the realistic assumption of weak irreversibility.

How does such a large relative volume of bistable parameter points in $\mathcal{H}^*$ become such a small relative volume in $\mathcal{H}$? To address this question, we examined more closely the values of $\epsilon_2$ and $\phi_0$ at which bistability occurs in the weakly irreversible case. Fig 7A shows the parameter

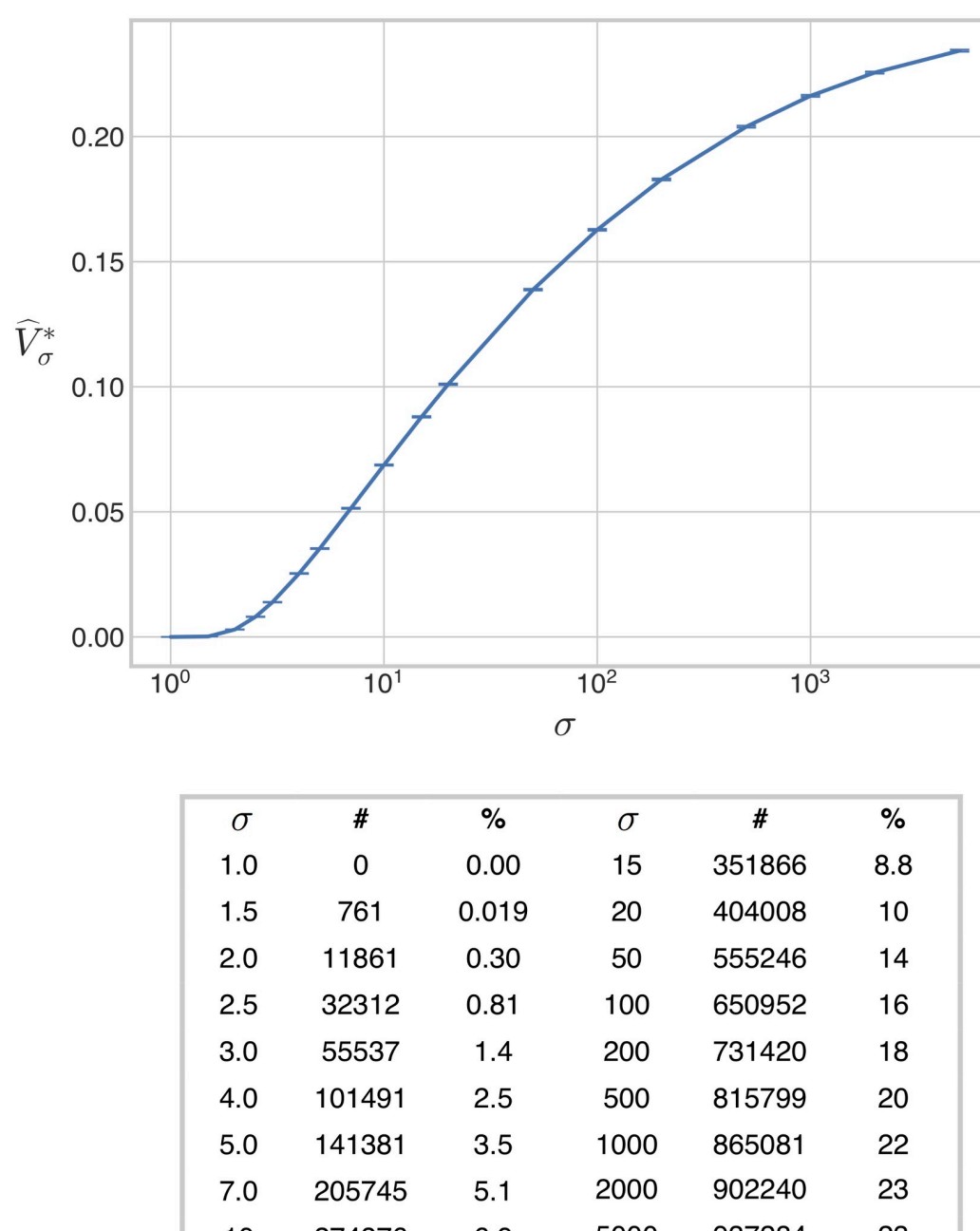

| $\sigma$ | # | % | $\sigma$ | # | % |
|---|---|---|---|---|---|
| 1.0 | 0 | 0.00 | 15 | 351866 | 8.8 |
| 1.5 | 761 | 0.019 | 20 | 404008 | 10 |
| 2.0 | 11861 | 0.30 | 50 | 555246 | 14 |
| 2.5 | 32312 | 0.81 | 100 | 650952 | 16 |
| 3.0 | 55537 | 1.4 | 200 | 731420 | 18 |
| 4.0 | 101491 | 2.5 | 500 | 815799 | 20 |
| 5.0 | 141381 | 3.5 | 1000 | 865081 | 22 |
| 7.0 | 205745 | 5.1 | 2000 | 902240 | 23 |
| 10 | 274876 | 6.9 | 5000 | 937284 | 23 |

**Fig 6. Volume of the bistable region under strong irreversibility.** The 6-dimensional volume of the bistable region, normalised as a proportion of the volume of the box $\mathcal{H}^*$, is plotted against the values of $\sigma$ in Eq 11, along with the additional values $\sigma = 1000, 2000, 5000$, for the case when the reactions $S_1 \overset{E}{\to} S_2$ and $S_1 \overset{F}{\to} S_0$ are strongly irreversible. The accompanying table lists the number of bistable points found for each value of $\sigma$, together with the percentage of the box $\mathcal{H}^*$ occupied by the bistable region. The error bars give 95% confidence intervals for each estimate (Materials and methods). The estimates have been joined by line segments.

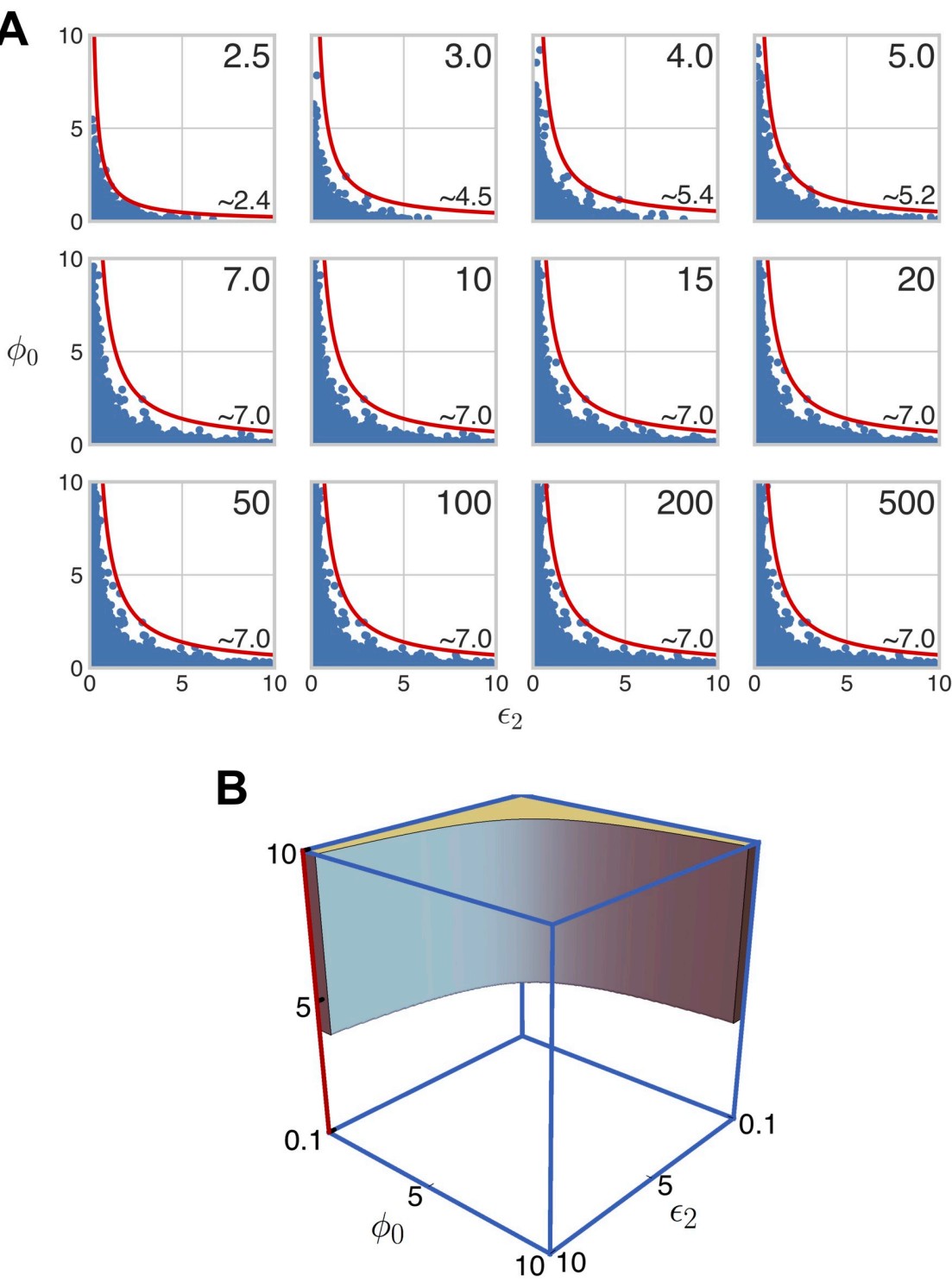

**Fig 7. Tradeoff in weak irreversibility for bistability.** A: The bistable regions under weak irreversibility, whose volumes are shown in Fig 3, are shown here projected onto the parameters $\theta_5 = \epsilon_2$ and $\theta_6 = \phi_0$ for the indicated values of $\sigma$ (top-right corner). In the bottom-right corner of each plot is the approximate value of the bound, $K$, on $\epsilon_2 \phi_0$, with the hyperbola $\epsilon_2 \phi_0 = K$ shown in red. B: A 3-dimensional schematic of the tradeoff between $\epsilon_2$ and $\phi_0$. The other six parameters are depicted as spanning a single dimension (the vertical axis). The 6-dimensional hypercube $\mathcal{H}^*$ is therefore a line segment along this vertical axis; the bistable region is shown occupying around 20% of this line segment (with respect to the logarithmic measure). The tradeoff between $\epsilon_2$ and $\phi_0$ yields a region that occupies a small volume in the 8-dimensional hypercube $\mathcal{H}$.

values, $\theta_5 = \epsilon_2$ and $\theta_6 = \phi_0$, for the bistable points used to calculate $\widehat{V}_\sigma$ in Fig 3. We immediately notice a striking tradeoff between the two parameters: bistability appears to be confined to the region in which

$$\epsilon_2 \phi_0 < K, \tag{19}$$

for some constant $K$ whose optimal value increases with $\sigma$. There appears, in other words, to be a tradeoff between bistability and product rebinding: the more of the latter, as given by $\epsilon_2\,\phi_0$ being large, the less of the former.

The constraint given by Eq 19 explains the dramatic decrease in volume from the strongly irreversible case in $\mathcal{H}^*$ to the weakly irreversible case in $\mathcal{H}$. The bistable region in the latter case is confined to the "thin" subregion of $\mathcal{H}$ in which $\epsilon_2$ and $\phi_0$ are not simultaneously large, as illustrated in Fig 7B.

## The bistable region outside $\mathcal{H}$

We have focused so far on the bistable region within boxes with sides [0.1, 10]. To assess how far these observations remain valid in larger regions of parameter space, we estimate here the volume of the bistable region in the boxes, $\mathcal{H}_p = [0.1^p, 10^p]^8$ for weak irreversibility, and $\mathcal{H}_p^* = [0.1^p, 10^p]^6$, for strong irreversibility, with $p$ running through the values $p = 2, 3, 4, 5$. As previously, we define $\mathcal{M}_{\sigma,p}, V_{\sigma,p}, \widehat{V}_{\sigma,p}$ to be, respectively, the bistable region in $\mathcal{H}_p$, its normalised volume and its estimated volume by Monte Carlo sampling and use an asterisk for the corresponding quantities $\mathcal{M}_{\sigma,p}^*, V_{\sigma,p}^*, \widehat{V}_{\sigma,p}^*$ for the bistable region in $\mathcal{H}_p^*$. In normalising the volumes, we note that the logarithmic volumes of the respective boxes are given by $V_{\mathcal{H}_p} = (2p)^8$ and $V_{\mathcal{H}_p^*} = (2p)^6$.

We generated samples of $10^6$ points by ILR sampling in each of the boxes $\mathcal{H}_2, \ldots, \mathcal{H}_5$ and $\mathcal{H}_2^*, \ldots, \mathcal{H}_5^*$ and ran Bertini and Paramotopy to find the bistable proportion of each sample at the 15 values of $\sigma$ listed in Eq 11, as well as six additional values of $\sigma$ between 1.0 and 1.5 which are discussed below (Eq 20). Despite the smaller sample sizes, we were able to find sufficiently many bistable points in each sample to obtain high-confidence volume estimates.

The results for weak irreversibility are shown in Fig 8. We found that, for each value of $p$, the volume of the bistable region restricted to $\mathcal{H}_p$ increases monotonically with $\sigma$, as we found previously for $p = 1$. The monotonic increase appears to be sigmoidal for each value of $p$, though saturation at large values of $\sigma$ becomes less evident as $p$ increases. We also found that the bistable proportion of $\mathcal{H}_p$ at a fixed value of $\sigma > 1$ increases monotonically with $p$, so that $V_{\sigma,a} < V_{\sigma,b}$ whenever $a < b$. This suggests that, as we expand the domain of sampling in parameter space, the rate at which we lose bistable points due to lower sampling density in $\mathcal{H}_a$ is exceeded by the rate at which we find bistable points in the complement, $\mathcal{H}_b - \mathcal{H}_a$. Moreover, we observed that, for all values of $\sigma$ greater than a threshold near $\sigma = 2.0$, the difference between volumes for successive values of $p$, $V_{\sigma,p+1} - V_{\sigma,p}$, decreases with $p$, suggesting the possibility that $V_{\sigma,p}$ increases as $p \to \infty$ towards an asymptotic functional dependence on $\sigma$, which is also monotonic and sigmoidal.

Strikingly, we also found that none of the points sampled in any of the boxes exhibits bistability at $\sigma = 1.0$, further supporting our prediction that a threshold for bistability exists. To find a more precise estimate of this threshold, we ran Bertini and Paramotopy on ILR samples acquired for $\mathcal{H}_2, \ldots, \mathcal{H}_5$ at each of the following six additional values of $\sigma$,

$$\sigma = 1.0078125, \ 1.015625, \ 1.03125, \ 1.0625, \ 1.125, \ 1.25. \tag{20}$$

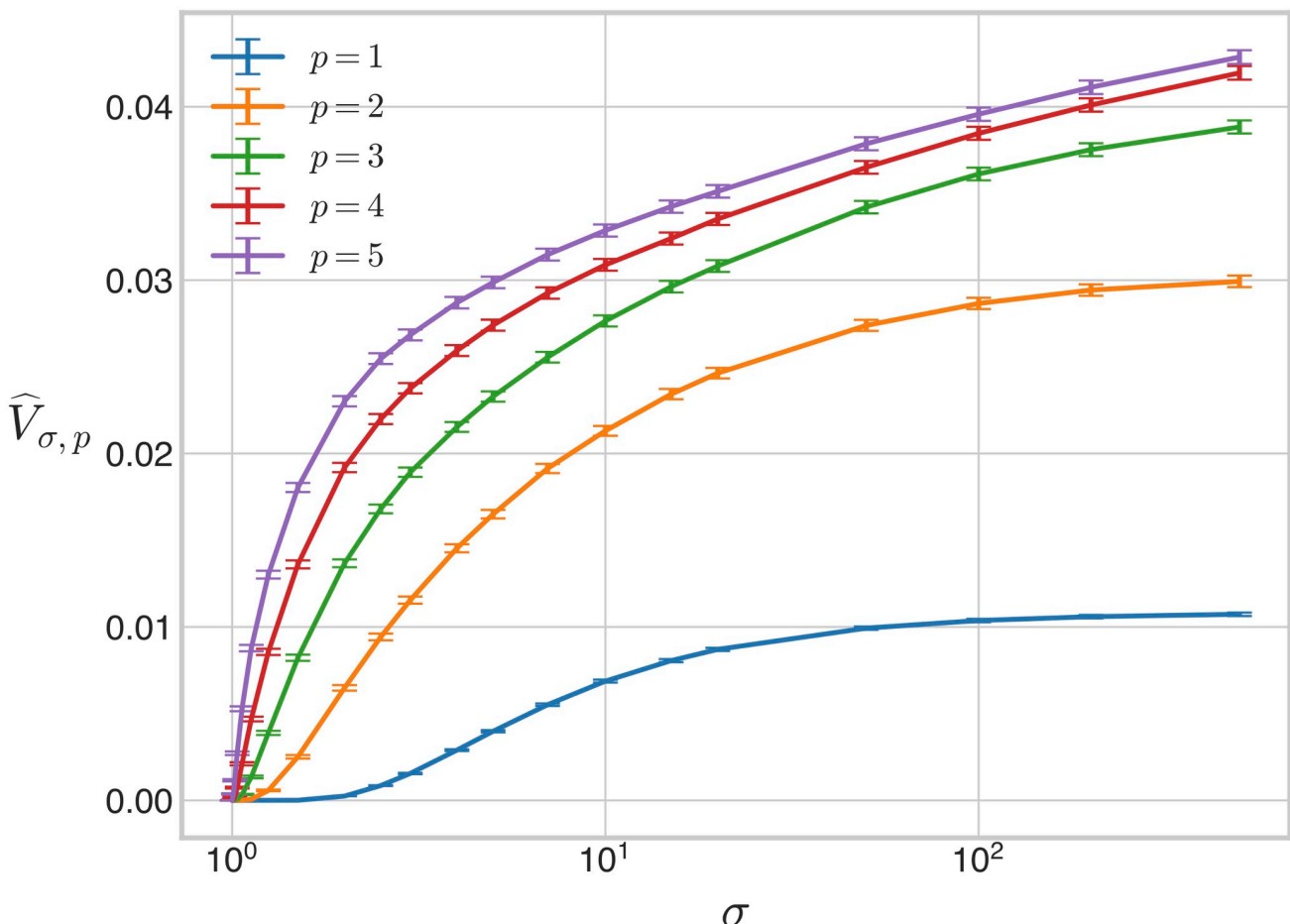

**Fig 8. Volume of the bistable region outside $\mathcal{H}$ under weak irreversibility.** The 8-dimensional volume, $\widehat{V}_{\sigma,p}$, of the bistable region under weak irreversibility, normalised as a proportion of the volume of the box, $\mathcal{H}_p = [0.1^p, 10^p]^8$, is plotted for the 21 values of $\sigma$ in Eqs 11 and 20, for $p = 2, 3, 4, 5$. The curve for $p = 1$ is taken from Fig 3. The error bars give 95% confidence intervals for each estimate (Materials and methods). The estimates have been joined by line segments.

We found bistable points at each of these values. For instance, we found 35 bistable points at $\sigma$ = 1.0078125 among the sample in $\mathcal{H}_4$, and as many as 372 bistable points at $\sigma$ = 1.0078125 among the sample in $\mathcal{H}_5$. This implies that, if a threshold for bistability exists, then it must be less than 1.0078125. Our failure to find any bistable points at $\sigma$ = 1.0 leads us to conjecture that the threshold is in fact exactly 1; we pose this as a conjecture for further study.

The results for strong irreversibility are shown in Fig 9. As in the weakly irreversible case, we found that, for each value of $p$, the volume of the bistable region in $\mathcal{H}_p^*$ increases monotonically with $\sigma$. We found no bistable points at $\sigma$ = 1 for any $p$, again consistent with the threshold conjecture.

However, we also observed three interesting differences between the weakly and strongly irreversible cases. First, although the asymptotic behaviour of the bistable volume as $\sigma \to \infty$ is difficult to resolve in the weakly irreversible case, the absence of saturation becomes conspicuous in the strongly irreversible case. The curves in Fig 9 suggest that under strong irreversibility the bistable volume increases more linearly with $\sigma$ when $\sigma$ is large. Second, we observed

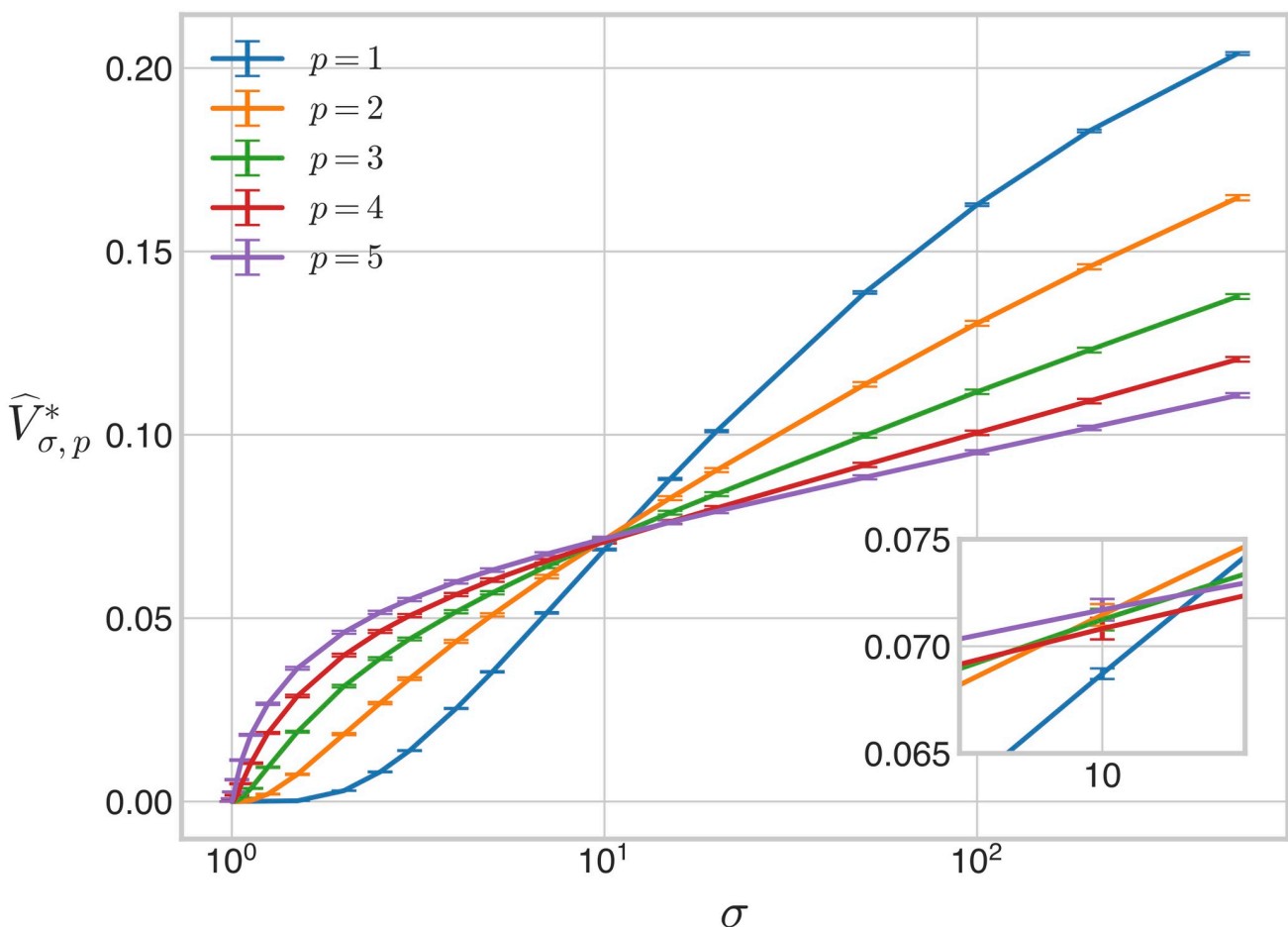

**Fig 9. Volume of the bistable region outside $\mathcal{H}^*$ under strong irreversibility.** The 6-dimensional volume, $\widehat{V}^*_{\sigma,p}$, of the bistable region under strong irreversibility, normalised as a proportion of the volume of the box, $\mathcal{H}^*_p = [0.1^p, 10^p]^6$, is plotted for the 21 values of $\sigma$ in Eqs 11 and 20, for $p = 2, 3, 4, 5$. The curve for $p = 1$ is taken from Fig 6. The error bars give 95% confidence intervals for each estimate (Materials and methods). The estimates have been joined by line segments. The inset shows the curves in the vicinity of $\sigma = 10$, at which point all five curves are close to intersecting. The 95% confidence interval for $\widehat{V}^*_{10,1}$ does not overlap with that for $\widehat{V}^*_{10,p}$ for $p = 2, 3, 4, 5$, indicating that the difference between the former and the latter is statistically significant at the 0.05 level.

that, as for $p = 1$, the bistable volume increases more rapidly with $\sigma$ under strong irreversibility than under weak irreversibility. However, this disparity becomes less prominent as $p$ increases: a roughly 20-fold difference between the bistable volumes at $p = 1$ ($V_{500,1} \approx 1.1\%$ and $V^*_{500,1} \approx 20\%$) decreases to a roughly 2.5-fold difference between the corresponding bistable volumes at $p = 5$ ($V_{500,5} \approx 4.3\%$ and $V^*_{500,5} \approx 11\%$).

Third, in contrast to the monotonically increasing dependence of $V_{\sigma,p}$ on $p$ for fixed $\sigma$, we find that $V^*_{\sigma,p}$ exhibits two qualitatively distinct patterns of dependence on $p$. For smaller values of $\sigma$, $V^*_{\sigma,p}$ monotonically increases with $p$; for larger values of $\sigma$, $V^*_{\sigma,p}$ monotonically decreases with $p$. The transition from the first to the second behaviour occurs close to $\sigma = 10$, at which value all five volume curves are close to intersecting. Among the five bistable volume estimates at $\sigma = 10$, $\widehat{V}^*_{10,1}$ appears to deviate the most from the others, and this difference is statistically

significant at the 0.05 level (note the non-overlapping 95% confidence intervals at $\sigma = 10$ in Fig 9). From this, it is difficult to say with confidence whether there is an interval of values for $\sigma$, near to or including $\sigma = 10$, in which $V^*_{\sigma,p}$ is non-monotonic in $p$. With that said, our analysis suggests that, for values of $\sigma$ outside such an interval, $V^*_{\sigma,p}$ depends monotonically on $p$ but is increasing for $\sigma$ smaller than the interval and decreasing for $\sigma$ greater than the interval. Here too, it seems that $V^*_{\sigma,p}$ may converge as $p \to \infty$ to an asymptotic functional dependence on $\sigma$ but in a seemingly different manner to the weakly irreversible case.

In summary, some of the observations made previously for the bistable region in $\mathcal{H}_1 = \mathcal{H}$ generalise to larger sampling domains in parameter space: the volume of the bistable region increases monotonically with $\sigma$ from a threshold, possibly toward a saturating volume in the weakly irreversible case. There is further compelling evidence for a bistability threshold which we conjecture to be at $\sigma = 1$. However, the size of the sampling domain affects bistable volumes in markedly different ways under weak and strong irreversibility.

## Discussion

Parametric robustness of a model—the maintenance of a property in the face of parametric change—offers a mathematical window onto the broader concept of biological robustness. We have approached this problem here through parameter geography, which offers an integrated, global view of the parametric region in which a property holds. We have been able to analyse the parameter geography of PTM systems with two sites but arbitrary enzymatic complexity and to estimate the size and shape of regions in 8-dimensional parameter space. We briefly summarise our most interesting findings and comment on their broader significance.

The approach we have taken has relied on randomly sampling parameter points and is therefore limited to features which have positive measure in the space being sampled. We have also conflated monostability and bistability with monostationarity and tristationarity, respectively, but merely as a convenience of language. With these caveats in mind, we found compelling evidence for a threshold in total substrate below which bistability does not exist. We found this both for weak irreversibility in all boxes $\mathcal{H}_p = [0.1^p, 10^p]^8$ and for strong irreversibility in all boxes $\mathcal{H}^*_p = [0.1^p, 10^p]^6$, with $p = 1, 2, 3, 4, 5$. In the light of other calculations not discussed here, we believe such a threshold may hold in greater generality and therefore put forward the following

**Conjecture**: Given a PTM system of the type studied in [17], consisting of a single substrate and multiple enzymes, $E_i$, each acting through any mechanism specified in the grammar of Eq 1, there exists a threshold level, $T(\{E_{i,\text{tot}}\}) > 0$, depending on the total amounts of each enzyme, such that the system shows no multistationarity if $S_{\text{tot}} < T(\{E_{i,\text{tot}}\})$.

For the two-site, distributive system studied here, in which we restricted attention to $E_{\text{tot}} = F_{\text{tot}}$, we found bistable points in $\mathcal{H}_4$ at $S_{\text{tot}}/E_{\text{tot}} = \sigma = 1.0078125$, which suggests that the threshold lies at $\sigma = 1$, so that $T(E_{\text{tot}}, F_{\text{tot}}) = E_{\text{tot}}$ when $E_{\text{tot}} = F_{\text{tot}}$. The conjecture above is consistent with recent work which showed that, when all enzymes follow the Michaelis-Menten reaction scheme, if $S_{\text{tot}} < E_{\text{tot}}$ or $S_{\text{tot}} < F_{\text{tot}}$, parameter values can be found at which there are multiple steady-states but such multistationarity does not exist if $S_{\text{tot}} > E_{\text{tot}}$ and $S_{\text{tot}} > F_{\text{tot}}$ [58, 89, 90].

Once past the threshold value, the bistable region acquires positive measure in the appropriate parameter space and thereby a volume. When $\sigma$ is large, the volume appears to saturate, although this is more pronounced for weak irreversibility and smaller boxes (Figs 8 and 9). However, there is a striking difference between the largest volumes reached under different enzymatic assumptions. The volume of the bistable region is substantially smaller under weak

irreversibility than under strong irreversibility. We were able to identify the reason for this as a constraint on enzyme parameters (Fig 7). This leads us to make another

**Conjecture**: For the two-site PTM system considered here, under the assumption of weak irreversibility, there exists a constant $K(\sigma)$, which increases with $\sigma$, such that there is no bistability when $\epsilon_2 \, \phi_0 > K$.

The parameters $\epsilon_2$ and $\phi_0$ depend on the ability of $S_2$ to rebind to the forward enzyme $E$, $\epsilon_2 = \kappa_{2,1}^E E_{\text{tot}}$, and the ability of $S_0$ to rebind to the reverse enzyme $F$, $\phi_0 = \kappa_{0,1}^F F_{\text{tot}}$, respectively (Eq 9). The above conjecture suggests there is a tradeoff between total rebinding and bistability: bistability is abrogated if both $E$ and $F$ exhibit high rebinding.

To our knowledge, there is no indication of such a relationship in the current literature. This is not surprising in view of the widespread fixation with the strongly irreversible Michaelis-Menten mechanism. As pointed out in the Introduction, one of our concerns has been to understand the impact of this unrealistic assumption on parametric robustness. Our results show that it can be very misleading. Under strong irreversibility, bistability appears robust, with the bistable region occupying 20% of $[0.1, 10]^6 = \mathcal{H}_1^*$ at $\sigma = 500$. Under weak irreversibility, bistability appears far rarer, occupying only 1.1% of $[0.1, 10]^8 = \mathcal{H}_1$ at $\sigma = 500$. The discrepancy becomes less marked for larger boxes, as noted above, but remains significant. A further difference between weak and strong irreversibility arises in the dependence of the bistable volume on the exponent, $p$, in the interval $[0.1^p, 10^p]$, which determines the size of the box: under weak irreversibility, volume increases monotonically with $p$ for each $\sigma$ (Fig 8); under strong irreversibility, volume increases monotonically for $\sigma < \sim 10$ and decreases monotonically for $\sigma > \sim 10$ (Fig 9). The reasons for this marked difference remain unclear but, here too, we see that unrealistic assumptions have significant consequences.

The implication of these findings is that a multisite PTM mechanism of the kind studied here is unlikely to act as a genuine biological memory or switch. It may require additional features to improve its robustness, such as scaffolding [91], to suggest only one possibility. Moreover, future studies will need to be more careful in their biochemical assumptions before making claims as to robustness.

The linear framework allows such care to be exercised by enabling realistic, complex enzyme mechanisms to be analysed at steady-state. This has not only brought out the differences between weak and strong irreversibility but also led to a potential explanation for the reduction in robustness with strong irreversibility, which lies in the tradeoff between rebinding and bistability expressed in the conjecture above. We hope these demonstrations will encourage less reliance on unrealistic assumptions and more focus on actual enzyme mechanisms.

Between low and high $\sigma$, the volume of the bistable region appears to increase monotonically: if $\sigma_1$ and $\sigma_2$ are taken from the list in Eq 11 and $\sigma_1 < \sigma_2$, then $\widehat{V}_{\sigma_1} < \widehat{V}_{\sigma_2}$. This holds for both weak (Figs 3 and 8) and strong (Figs 6 and 9) irreversibility. It is natural to expect that such monotonicity arises because the bistable region itself increases with $\sigma$, so that $\mathcal{M}_{\sigma_1} \subset \mathcal{M}_{\sigma_2}$. However, this is not the case: parameter points can move back and forth between monostability and bistability (Fig 5). The pattern of blinking, in which a bistable point becomes monostable, exhibits several interesting features (Table 2), which suggest that blinking may be restricted to the boundary of the bistable region.

These findings are perplexing and suggest surprising complexity in how the steady-state manifold is positioned relative to the hypersurfaces defined by conservation of total substrate and total enzymes. One way to focus on the problem is to ask whether or not the volume of the bistable region, $V_\sigma$, is a monotonic function of $\sigma$. If it is monotonic, this makes it difficult to account for blinking points, whose disappearance from $\mathcal{M}_\sigma$ must then be compensated by the appearance of other bistable points. If it is not monotonic, this makes it difficult to account for

the monotonicity of the estimated volume, $\widehat{V}_\sigma$, as shown in Fig 3. It is, of course, possible that departure from monotonicity of $V_\sigma$ does occur but at a scale that is too small to be observed by random sampling. If so, it becomes an interesting problem to determine the scale of these volume fluctuations. Such challenges lead us to pose the following

**Question**: How does $V_\sigma$ depend on $\sigma$? If the relationship is monotonic, how is blinking compensated? If it is not monotonic, on what scale does it depart from monotonicity so that the estimated volume, $\widehat{V}_\sigma$, depends monotonically on $\sigma$?

The volume of the bistable region indicates its size but not its shape. This becomes a subtle problem in high dimensions. Even in two dimensions, it is evident that local variation in shape can greatly compromise robustness (Figs 1B and 4B). It is interesting, therefore, that the bistable region appears to be well behaved for the simplest shape measures. It is both connected (Table 1) and nearly convex, with a visibility ratio over 95% for all values of $\sigma$ (Fig 4A). As far as these limited measures of shape are concerned, the bistable region exhibits good robustness. To know more, it would be necessary to use methods like persistent homology, which give access to higher-dimensional topological invariants [92, 93]. While this lies beyond the scope of the present paper, the datasets arising from our analysis are available for others to explore this question (S1 File; see also S1 Appendix). It would be of considerable interest to have an estimate of the topological type of the bistable region, which could stimulate further mathematical conjectures like those above.

Exploring the conjectures and question above presents an interesting challenge. Approaches based on chemical reaction network theory [56, 58–62] may be useful for clarifying whether our conjectures hold for particular choices of enzyme mechanism but they may need to be adapted to rise above the details of such mechanisms. Discriminant locus approaches have been used to identify parameter regions for multistationarity but have so far been limited to parameter spaces of low dimension [67].

The analysis undertaken here, while based on numerical calculations and statistical analysis of data, has led to precise mathematical conjectures and questions about parameter geography. This has been possible by bringing together two advances: the linear framework, which enables steady-state reduction of PTM systems to polynomial equations, and numerical algebraic geometry, in the form of Bertini and related software tools, which allow efficient sampling of high-dimensional parameter spaces. This suggests that a new kind of exploratory mathematics is becoming feasible to study those biological systems which can be treated in this way, through timescale separation and polynomial specification. It may now be possible to accommodate more of the molecular complexity that is found in biology, as we have done here for enzyme mechanisms, but to also elicit mathematical insights which rise above that complexity, as described in the conjectures above. Perhaps a theory of parameter geography may eventually crystallise from studies of this kind.

## Materials and methods

### Derivation of Eq 10

We use the linear framework to show that the steady-state of the two-site PTM system described in Fig 2 is given by the solutions of the two polynomial equations given in Eq 10.

Each of the four modifications in the system is of the form, $S_i \xrightarrow{X} S_j$, where the indices $i, j \in \{0, 1, 2\}$ enumerate the modforms and $X \in \{E, F\}$ is an enzyme (Fig 2). We assume that the aggregate mechanism $S_i \xrightarrow{X} S_j$ is described by a graph of elementary reactions in the grammar in Eq 1, with a finite number of enzyme-substrate intermediate complexes. We further assume that the sets of enzyme-substrate complexes involved in distinct enzyme mechanisms are

disjoint. Then, by Proposition 1 in [17], the steady-state concentration of any enzyme-substrate complex, $Y_\ell$, is given by

$$[Y_\ell] = (\mu_{i,\ell}^X [S_i] + \mu_{j,\ell}^X [S_j])[X] \, ,$$

where $\mu_{i,\ell}^X$ and $\mu_{j,\ell}^X$ are <u>reciprocal generalised Michaelis-Menten constants</u> (rgMMCs). The linear framework guarantees that these parameters are positive for each $Y_\ell$, unless $S_i \xrightarrow{X} S_j$ is strongly irreversible, in which case there is no flux from $S_j$ to any of the enzyme-substrate complexes, and so $\mu_{j,\ell}^X = 0$ for every $Y_\ell$ involved in the mechanism for $S_i \xrightarrow{X} S_j$.

We may aggregate the rgMMCs by summing the concentrations of the enzyme-substrate complexes over each modification, to yield Eq 2,

$$\sum_{S_0 \xrightarrow{E} S_1} [Y_*] \quad = (\kappa_{0,1}^E [S_0] + \kappa_{1,0}^E [S_1])[E]$$

$$\sum_{S_1 \xrightarrow{E} S_2} [Y_*] \quad = (\kappa_{1,2}^E [S_1] + \kappa_{2,1}^E [S_2])[E]$$

$$\sum_{S_2 \xrightarrow{F} S_1} [Y_*] \quad = (\kappa_{2,1}^F [S_2] + \kappa_{1,2}^F [S_1])[F]$$

$$\sum_{S_1 \xrightarrow{F} S_0} [Y_*] \quad = (\kappa_{1,0}^F [S_1] + \kappa_{0,1}^F [S_0])[F] \, ,$$

where

$$\kappa_{0,1}^E = \sum_{S_0 \xrightarrow{E} S_1} \mu_{0,*}^E \qquad\qquad \kappa_{0,1}^F = \sum_{S_1 \xrightarrow{F} S_0} \mu_{0,*}^F$$

$$\kappa_{1,0}^E = \sum_{S_0 \xrightarrow{E} S_1} \mu_{1,*}^E \qquad\qquad \kappa_{1,0}^F = \sum_{S_1 \xrightarrow{F} S_0} \mu_{1,*}^F$$

$$\kappa_{1,2}^E = \sum_{S_1 \xrightarrow{E} S_2} \mu_{1,*}^E \qquad\qquad \kappa_{1,2}^F = \sum_{S_2 \xrightarrow{F} S_1} \mu_{1,*}^F$$

$$\kappa_{2,1}^E = \sum_{S_1 \xrightarrow{E} S_2} \mu_{2,*}^E \qquad\qquad \kappa_{2,1}^F = \sum_{S_2 \xrightarrow{F} S_1} \mu_{2,*}^F$$

are the reciprocal total generalised Michaelis-Menten constants (rtgMMCs). When the four mechanisms are each strongly irreversible, we have

$$\kappa_{1,0}^E = \kappa_{2,1}^E = \kappa_{0,1}^F = \kappa_{1,2}^F = 0 \, ,$$

since the rgMMCs in the corresponding sums are identically zero.

We may further aggregate the rtgMMCs as follows,

$$\sum_{S_0 \xrightarrow{E} S_1} [Y_*] + \sum_{S_1 \xrightarrow{E} S_2} [Y_*] \quad = (\kappa_0^E [S_0] + \kappa_1^E [S_1] + \kappa_2^E [S_2])[E]$$

$$\sum_{S_2 \xrightarrow{F} S_1} [Y_*] + \sum_{S_1 \xrightarrow{F} S_0} [Y_*] \quad = (\kappa_0^F [S_0] + \kappa_1^F [S_1] + \kappa_2^F [S_2])[F] \, ,$$

(21)

where

$$\kappa_0^X = \kappa_{0,1}^X, \qquad \kappa_1^X = \kappa_{1,0}^X + \kappa_{1,2}^X, \qquad \kappa_2^X = \kappa_{2,1}^X \qquad \text{for } X = E, F.$$

When the four mechanisms are strongly irreversible, we have $\kappa_2^E = \kappa_0^F = 0$.

Note that Eq 21 and the enzyme conservation laws, Eq 8, imply that $[E] = [F] = 0$ if, and only if, $E_{tot} = F_{tot} = 0$, which we assume is not the case. Therefore, $[E]$ and $[F]$ must be positive. (This is relevant to the steps that follow in which we divide by $[E]$ or $[F]$.)

Proposition 1 in [17] implies that under mass action kinetics, the dynamics of the substrate mod-forms may be written as follows,

$$
\begin{aligned}
\frac{d[S_0]}{dt} &= (c_{1,0}^E[E] + c_{1,0}^F[F])[S_1] - (c_{0,1}^E[E] + c_{0,1}^F[F])[S_0] \\[2mm]
\frac{d[S_1]}{dt} &= (c_{0,1}^E[E] + c_{0,1}^F[F])[S_0] + (c_{2,1}^E[E] + c_{2,1}^F[F])[S_2] \\[2mm]
&\quad - (c_{1,0}^E[E] + c_{1,0}^F[F] + c_{1,2}^E[E] + c_{1,2}^F[F])[S_1] \\[2mm]
\frac{d[S_2]}{dt} &= (c_{1,2}^E[E] + c_{1,2}^F[F])[S_1] - (c_{2,1}^E[E] + c_{2,1}^F[F])[S_2] \,,
\end{aligned}
\tag{22}
$$

where $c_{i,j}^X$ are the tgCEs. Assuming the forward-modification tgCEs are positive, $c_{0,1}^E,\ c_{1,2}^E,\ c_{1,0}^F,\ c_{2,1}^F > 0$, and the reverse-demodification tgCEs are zero, $c_{1,0}^E = c_{2,1}^E = c_{0,1}^F = c_{1,2}^F = 0$, we have,

$$
c_{0,1}^E[E][S_0] = c_{1,0}^F[F][S_1] \qquad \text{and} \qquad c_{1,2}^E[E][S_1] = c_{2,1}^F[F][S_2]
$$

at steady-state. This yields Eq 5,

$$
[S_0] = \alpha\left(\frac{[F]}{[E]}\right)[S_1] \qquad \text{and} \qquad [S_2] = \beta\left(\frac{[E]}{[F]}\right)[S_1] \,,
$$

where $\alpha$ and $\beta$ are as defined in Eq 6,

$$
\alpha = \frac{c_{1,0}^F}{c_{0,1}^E} \qquad \text{and} \qquad \beta = \frac{c_{1,2}^E}{c_{2,1}^F} \,.
$$

Under our assumptions, $\alpha, \beta, [E]$, and $[F]$ are all positive. Hence, Eq 5 implies that, if any one of the modform concentrations is zero, then all of them are. This implies in turn, via Eq 21 and the substrate conservation law Eq 7, that $S_{tot}$ is zero. Therefore, the modform and enzyme concentrations are positive at steady-state, as long as we assume that the following are positive: the conserved quantities $S_{tot}, E_{tot}, F_{tot}$; any one of the rtgMMCs for $E$, $\kappa_0^E,\ \kappa_1^E,\ \kappa_2^E$; any one of the rtgMMCs for $F$, $\kappa_0^F,\ \kappa_1^F,\ \kappa_2^F$; and all of the forward-modification tgCEs $c_{0,1}^E,\ c_{1,2}^E,\ c_{2,1}^F,\ c_{1,0}^F$. The strongly and weakly irreversible cases both satisfy these assumptions.

We can combine Eqs 5 and 21 to write

$$
\begin{aligned}
[S_0] + [S_1] + [S_2] &= \left(\alpha\frac{[F]}{[E]} + 1 + \beta\frac{[E]}{[F]}\right)[S_1] \\[3mm]
\sum_{S_0 \xrightarrow{E} S_1}[Y_*] + \sum_{S_1 \xrightarrow{E} S_2}[Y_*] &= \left(\kappa_0^E\alpha\frac{[F]}{[E]} + \kappa_1^E + \kappa_2^E\beta\frac{[E]}{[F]}\right)[S_1][E] \\[3mm]
\sum_{S_2 \xrightarrow{F} S_1}[Y_*] + \sum_{S_1 \xrightarrow{F} S_0}[Y_*] &= \left(\kappa_0^F\alpha\frac{[F]}{[E]} + \kappa_1^F + \kappa_2^F\beta\frac{[E]}{[F]}\right)[S_1][F] \,.
\end{aligned}
\tag{23}
$$

Define the rational functions

$$
\begin{aligned}
\psi_1([E],[F]) \quad &= \alpha \frac{[F]}{[E]} + 1 + \beta \frac{[E]}{[F]} \\
\psi_2([E],[F]) \quad &= E_{\text{tot}} \left( \kappa_0^E \alpha \frac{[F]}{[E]} + \kappa_1^E + \kappa_2^E \beta \frac{[E]}{[F]} \right) \\
\psi_3([E],[F]) \quad &= F_{\text{tot}} \left( \kappa_0^F \alpha \frac{[F]}{[E]} + \kappa_1^F + \kappa_2^F \beta \frac{[E]}{[F]} \right),
\end{aligned}
$$

and substitute them into Eq 23:

$$
\begin{aligned}
[S_0] + [S_1] + [S_2] \quad &= [S_1]\psi_1 \\
\sum_{S_0 \xrightarrow{E} S_1} [Y_*] + \sum_{S_1 \xrightarrow{E} S_2} [Y_*] \quad &= \frac{[S_1][E]}{E_{\text{tot}}} \psi_2 \\
\sum_{S_2 \xrightarrow{F} S_1} [Y_*] + \sum_{S_1 \xrightarrow{F} S_0} [Y_*] \quad &= \frac{[S_1][F]}{F_{\text{tot}}} \psi_3.
\end{aligned} \tag{24}
$$

Substituting Eq 24 into the substrate conservation law, Eq 7, gives

$$
S_{\text{tot}} = \left( \psi_1 + \frac{[E]}{E_{\text{tot}}}\psi_2 + \frac{[F]}{F_{\text{tot}}}\psi_3 \right)[S_1] ; \tag{25}
$$

and substituting Eq 24 into the enzyme conservation laws, Eq 8, gives

$$
E_{\text{tot}} = \left( 1 + \frac{[S_1]}{E_{\text{tot}}}\psi_2 \right)[E] \qquad \text{and} \qquad F_{\text{tot}} = \left( 1 + \frac{[S_1]}{F_{\text{tot}}}\psi_3 \right)[F]. \tag{26}
$$

These substitutions eliminate all of the state variables except for $[S_1]$, $[E]$, and $[F]$. We then eliminate $[S_1]$ by substituting Eq 25 into Eq 26, to get

$$
\begin{aligned}
E_{\text{tot}} \quad &= \left( 1 + \frac{S_{\text{tot}}/E_{\text{tot}}}{\psi_1 + ([E]/E_{\text{tot}})\psi_2 + ([F]/F_{\text{tot}})\psi_3} \psi_2 \right)[E] \\
F_{\text{tot}} \quad &= \left( 1 + \frac{S_{\text{tot}}/F_{\text{tot}}}{\psi_1 + ([E]/E_{\text{tot}})\psi_2 + ([F]/F_{\text{tot}})\psi_3} \psi_3 \right)[F].
\end{aligned} \tag{27}
$$

Finally, we substitute the non-dimensional quantities,

$$
\begin{aligned}
u \quad &= \frac{[E]}{E_{\text{tot}}} , \quad v = \frac{[F]}{F_{\text{tot}}} , \quad \epsilon_0 = \kappa_0^E E_{\text{tot}} , \quad \epsilon_1 = \kappa_1^E E_{\text{tot}} , \quad \epsilon_2 = \kappa_2^E E_{\text{tot}} , \\
\sigma \quad &= \frac{S_{\text{tot}}}{E_{\text{tot}}} , \quad \lambda = \frac{S_{\text{tot}}}{F_{\text{tot}}} , \quad \phi_0 = \kappa_0^F F_{\text{tot}} , \quad \phi_1 = \kappa_1^F F_{\text{tot}} , \quad \phi_2 = \kappa_2^F F_{\text{tot}} ,
\end{aligned}
$$

into $\psi_1$, $\psi_2$, and $\psi_3$ to get

$$
\begin{aligned}
\psi_1(u, v) &= \frac{\alpha}{\zeta}\frac{v}{u} + 1 + \beta\zeta\frac{u}{v} \\
\psi_2(u, v) &= \frac{\epsilon_0\alpha}{\zeta}\frac{v}{u} + \epsilon_1 + \epsilon_2\beta\zeta\frac{u}{v} \\
\psi_3(u, v) &= \frac{\phi_0\alpha}{\zeta}\frac{v}{u} + \phi_1 + \phi_2\beta\zeta\frac{u}{v}\,,
\end{aligned}
$$

and into Eq 27 to get

$$
\begin{aligned}
\frac{1}{u} &= 1 + \frac{\sigma}{\psi_1 + u\psi_2 + v\psi_3}\psi_2 \\
\frac{1}{v} &= 1 + \frac{\lambda}{\psi_1 + u\psi_2 + v\psi_3}\psi_3\,.
\end{aligned}
\tag{28}
$$

Cross-multiplying Eq 28 gives

$$
\begin{aligned}
(1-u)(\psi_1 + u\psi_2 + v\psi_3) - \sigma u\psi_2 &= 0 \\
(1-v)(\psi_1 + u\psi_2 + v\psi_3) - \lambda v\psi_3 &= 0.
\end{aligned}
\tag{29}
$$

Finally, we note that the two left-hand-side expressions in Eq 29 are not polynomials in $u$ and $v$, since $\psi_1$, $\psi_2$, and $\psi_3$ include terms with both $u/v$ and $v/u$. So we multiply both sides by $\zeta uv$ to get,

$$
\begin{aligned}
\Phi_1(u, v) &= \zeta uv((1-u)(\psi_1 + u\psi_2 + v\psi_3) - \sigma u\psi_2) = 0 \\
\Phi_2(u, v) &= \zeta uv((1-v)(\psi_1 + u\psi_2 + v\psi_3) - \lambda v\psi_3) = 0\,.
\end{aligned}
\tag{30}
$$

One can check that expanding Eq 30 and substituting in the expressions for $\psi_1$, $\psi_2$, and $\psi_3$ gives Eq 10.

## Solving Eq 10 with Bertini and Paramotopy

We briefly describe how homotopy continuation works and discuss the practical issues we encountered in using Bertini and Paramotopy. For more complete details, see [77]. A complete description of our workflow, along with details on the supplemental code and datasets, is given in S1 Appendix.

**Homotopy continuation with Bertini.**   Consider a system of $n$ polynomial equations in $n$ variables,

$$
f(x) = f(x_1, \ldots, x_n) = \begin{bmatrix} f_1(x_1, \ldots, x_n) \\ f_2(x_1, \ldots, x_n) \\ \vdots \\ f_n(x_1, \ldots, x_n) \end{bmatrix} = 0\,.
$$

We are interested in solutions of such systems which consist of finitely many isolated points. Homotopy continuation uses two steps. First, another system of $n$ polynomial equations, the start system, $g(x) = 0$, is selected, whose set of solutions can be easily computed. Second, a

continuous map $H : \mathbb{C}^n \times [0, 1] \to \mathbb{C}^n$ is defined so as to give a <u>homotopy</u> between $f$ and $g$,

$$H(x_1, \ldots, x_n, 1) = g(x_1, \ldots, x_n) \qquad \text{and} \qquad H(x_1, \ldots, x_n, 0) = f(x_1, \ldots, x_n).$$

The idea is that, as $t$ is changed from 1 to 0, $H(x_1, \ldots, x_n, t) = 0$ traces a continuous deformation, or <u>path</u>, from each of the known solutions of $g(x_1, \&, x_n) = 0$ to the desired solutions of $f(x_1, \ldots, x_n) = 0$.

Bertini's default choice of homotopy $H$ is the <u>total-degree homotopy</u>, given by,

$$H(x_1, \ldots, x_n, t) = (1 - t)f(x_1, \ldots, x_n) + \gamma\, t\, g(x_1, \ldots, x_n),$$

where $\gamma \neq 0$ is a random complex number and $g$ is a <u>total-degree start system</u>, which is one that has the maximal number of finite isolated solutions given by Bézout's Theorem. For example, we may set $g_i$, for $i = 1, \ldots, n$, to the polynomial [77]

$$g_i(x_1, \ldots, x_n) = x_i^{d_i} - 1,$$

where $d_i$ is the degree of $f_i$. We found this choice of start system and homotopy to be adequate for our analysis.

The homotopy $H$ defines a complex-valued path from each solution of $g(x_1, \ldots, x_n) = 0$ to a solution of $f(x_1, \ldots, x_n) = 0$. Bertini tracks these paths using numerical <u>predictor-corrector methods</u>. More sophisticated algorithms, called <u>endgames</u>, are used to track the paths with enhanced precision once $t$ is close to zero. The use of projective coordinates, which introduce additional points at infinity, allows the reliable tracking of divergent paths to arbitrary precision.

A solution $x^*$ is <u>non-singular</u> if it has multiplicity 1; otherwise, the solution is <u>singular</u>. Bertini determines the endpoint $x^*$ of a homotopy path to be singular if the condition number of the Jacobian matrix of $f$ at $x^*$ is large. A non-singular solution, if reasonably accurate, can be <u>sharpened</u> to arbitrarily many digits by performing additional post-endgame iterations of Newton's method. This provides a rapid alternative to re-tracking the homotopy with more stringent Bertini settings to obtain highly accurate non-singular solutions. It also guarantees a desired level of accuracy irrespective of the path-tracking behaviour. However, this method is not useful for obtaining highly accurate singular solutions, near which Newton's method can converge more slowly, or not at all [77]. Bertini incorporates a large number of customisable settings which control its path-tracking behaviour; for a comprehensive discussion see [77].

**Parameter homotopy continuation with Paramotopy.**   The polynomial systems of interest to us contain parameters and can be written as $f(x; \theta) = 0$, where $x = (x_1, \ldots, x_n)$ and $\theta = (\theta_1, \ldots \theta_m) \in \mathbb{C}^m$. We typically have a set of parameter points, $\theta^{(1)}, \ldots, \theta^{(N)} \in \mathbb{C}^m$, at which solutions of $f(x; \theta^{(i)}) = 0$ are required. Paramatopy is a software package built on Bertini which allows such parameterised polynomial systems to be solved efficiently in parallel through homotopies in parameter space. It uses the following two-step process.

**Step 1**. Randomly sample a parameter point, $\theta^* \in \mathbb{C}^m$, and find all the isolated solutions of $f(x; \theta^*) = 0$ using homotopy continuation in Bertini.

**Step 2**. For each $j = 1, \ldots, N$, solve the system $f(x; \theta^{(j)}) = 0$ by tracking in parallel the solutions of the parameter homotopy,

$$f(x; p(t)) = 0, \qquad \text{where} \qquad p(t) = (1 - t)\theta^{(j)} + t\, \theta^*.$$

Paramotopy exploits the fact that a parameterised polynomial system has the same number of non-singular solutions at any parameter point outside a set of measure zero [77]. Therefore, if the system has $k$ non-singular solutions, and $d \geq k$ is the maximal number of finite isolated solutions given by Bézout's Theorem, then Paramotopy would track $d$ paths during Step 1 and

$kN$ paths during Step 2. This can offer significant computational savings in comparison with running Bertini $N$ times, once for each parameter point $\theta^{(j)}$, which would require tracking a total of $dN$ paths.

As noted in the main text, we have $d = 16$ for Eq 10. In the weakly irreversible case, we found $k = 7$ "proper" (meaning nonzero, non-singular and non-projectively infinite) complex solutions at each parameter point. In the strongly irreversible case, we found $k = 5$ proper solutions.

Despite having a wide range of path-tracking methods, Bertini may still fail to track a path to $t = 0$ if numerical difficulties arise. Paramotopy collects all such instances of early path track-ing termination, called path failures, and can re-track these failed homotopies with a new Step 1 parameter point. This can be repeated as many times as necessary until all path failures are resolved.

**Use of cluster computing.** We ran Bertini and Paramotopy on the Orchestra and O2 high-performance computing clusters at Harvard Medical School. The vast majority of the Paramotopy runs were performed on Orchestra, with the Step 2 path-tracking performed in parallel over 10–20 cores per batch of $2.5 \times 10^5$ parameter points; the remaining computations were performed on O2, which succeeded Orchestra in March 2018. O2 currently consists of more than 11000 cores across 300 Intel Xeon x86 multi-core processors of various specifications.

The Paramotopy runs are generally time-consuming: solving the system in Eq 10 on a batch of $2.5 \times 10^5$ parameter points in a single Paramotopy run usually requires a runtime of several hours. The exact runtime depends on the values of the parameters and conserved quantities, the number of cores used, and details of the underlying numerics, such as the choice of Step 1 parameter point $\theta^*$ (see above) and the Bertini settings (see below). See S1 Appendix for details.

**Classifying solutions and numerical settings.** Bertini and Paramotopy incorporates various tunable settings for distinguishing between zero and nonzero endpoints (`ImagThreshold`) and between finite and projectively infinite endpoints (`EndpointFiniteThreshold`). However, enforcing a fixed threshold can lead to mis-leading conclusions, as the accuracy of a solution obtained through parameter homotopy continuation depends on many factors. We therefore used a different strategy for distin-guishing between zero, nonzero finite, and projectively infinite values for each solution obtained via Paramotopy.

A reported numerical solution coordinate, $x^* = a^* + ib^* \in \mathbb{C}$, was categorised as one of the following, depending on three positive thresholds, $T_{\mathrm{zmin}}$, $T_{\mathrm{zmax}}$, and $T_\infty$, chosen such that $0 < T_{\mathrm{zmin}} \geq T_{\mathrm{zmax}} \ll 1 \ll T_\infty$:

$$x^* \text{ is } \begin{cases} \underline{\text{small}} & \text{if } |a^*|, \ |b^*| < T_{\mathrm{zmax}} \\ \underline{\text{ambiguous}} & \text{if } T_{\mathrm{zmin}} < |b^*| < T_{\mathrm{zmax}} \\ \underline{\text{infinite}} & \text{if } |a^*| > T_\infty \text{ or } |b^*| > T_\infty \\ \underline{\text{nonzero real}} & \text{if } T_{\mathrm{zmax}} < |a^*| < T_\infty \text{ and } |b^*| < T_{\mathrm{zmin}} \\ \underline{\text{nonzero non-real}} & \text{otherwise.} \end{cases} \tag{31}$$

A reported solution, $(u^*, v^*) \in \mathbb{C}^2$, was determined to be proper if (1) $(u^*, v^*)$ was reported by Paramotopy as non-singular, and (2) both $u$ and $v^*$ were categorised as nonzero real or non-zero non-real.

We also determined a reported solution coordinate, $x^* = a^* + ib^*$, to be insufficiently pre-cise if Paramotopy specified either $a^*$ or $b^*$ with fewer than $T_d$ digits, for some positive integer

**Table 3. First-pass Bertini settings.** The principal settings used for the computational analysis are listed here. Every first-pass Paramotopy run was performed using these Bertini settings; all other settings were set to their default values [77]. The column title PFR refers to "path failure resolution" and this column gives Bertini setting values used for the first iteration of this process; the Bertini settings were modulated appropriately for subsequent iterations. A complete list of all Bertini settings used for every Paramotopy run and re-run, as well as all path failure resolution iterations, is given in S1 Dataset, with further details in S1 Appendix.

| Setting | Step 1 | Step 2 | PFR |
|---|---|---|---|
| CondNumThreshold | $10^8$ | $10^{14}$ | $10^{14}$ |
| FinalTol | $10^{-11}$ | $10^{-11}$ | $10^{-8}$ |
| MaxNorm | $10^5$ | $10^5$ | $10^8$ |
| MaxNumberSteps | 10000 | 10000 | 20000 |
| SecurityLevel | 0 | 1 | 1 |
| SharpenDigits | 0 | 25 | 0 |
| SharpenOnly | 0 | 1 | 0 |
| TrackTolBeforeEG | $10^{-5}$ | $10^{-5}$ | $10^{-6}$ |
| TrackTolDuringEG | $10^{-6}$ | $10^{-6}$ | $10^{-7}$ |
| UserHomotopy | 0 | 2 | 2 |

$T_d$. By using Bertini's sharpening module, most non-singular solutions were specified with the desired precision but insufficiently precise solutions could occasionally arise in one of two ways. On the one hand, a path failure in Step 2 that was not subsequently resolved with sufficiently stringent Bertini settings occasionally manifested as an insufficiently precise solution. On the other hand, an ill-conditioned Jacobian matrix (whose condition number exceeds an internal threshold) was occasionally encountered during sharpening for a small minority of solutions—many of which were at parameter points sampled from $\mathcal{H}_4^*$ or $\mathcal{H}_5^*$—at which point Bertini terminated sharpening prematurely. In view of these numerical issues, we set $T_d$ to slightly (usually five) fewer digits than the value of `SharpenDigits` (Table 3 and below) for the corresponding Paramotopy run.

Using the classification in Eq 31, we were able to find the generic number of proper solutions to Eq 10 with sufficient precision—seven under weak irreversibility and five under strong irreversibility—for the majority of parameter points on the first attempt, using the first-pass Bertini settings in Table 3 and the threshold values $T_{zmin} = 10^{-25}$, $T_{zmax} = 10^{-10}$, $T_\infty = 10^8$, and $T_d = 20$. However, each such first-pass Paramotopy run resulted in between $\sim 100$ and $\sim 10^5$ parameter points with at least one questionable solution (i.e., a solution reported by Paramotopy as singular, or a solution with at least one small, ambiguous, infinite, or insufficiently precise coordinate). We therefore re-ran Paramotopy on all such points with a new choice of Step 1 point and more stringent Bertini settings. Upon obtaining a new solution set for each of these parameter points, we repeated this process—collecting all parameter points with at least one questionable solution and re-running Paramotopy on these points with increasingly stringent Bertini settings—until we identified the generic number of proper solutions for each point in the entire sample. We collected together a final set of proper solutions for each point in the sample, with which we performed all downstream analyses. The list of all Paramotopy runs and re-runs, and the Bertini settings used for each, are given in S1 Dataset; see S1 Appendix for further details.

We found by manual exploration that setting $T_{zmin} = 10^{-25}$, $T_{zmax} = 10^{-10}$, and $T_\infty = 10^8$ was appropriate for most Paramotopy runs. For a small subset of Paramotopy re-runs, we found that alternative values for the thresholds were more appropriate, based on seeing repeated convergence to questionable values despite stringent Bertini settings. For instance, we found that re-running Paramotopy on certain parameter points with more stringent Bertini settings yielded solutions with imaginary parts with absolute value less than $10^{-10}$ but greater

than $10^{-11}$, so that $T_{zmax} = 10^{-11}$ is a more appropriate choice. Likewise, certain parameter points also exhibited seemingly finite solutions with absolute value greater than $10^8$, necessitating the use of larger values for $T_\infty$.

As mentioned above, we set $T_d$ to (usually) five fewer digits than the value of `SharpenDigits` for the corresponding run or re-run. For instance, each first-pass Paramotopy run was performed with `SharpenDigits` set to 25 (Table 3), and we set $T_d = 20$ when classifying the solutions reported from these runs. Subsequent re-runs were performed with incrementally increasing values for `SharpenDigits` (S1 Dataset), with $T_d$ also increasing proportionately.

The list of values used for $T_{zmin}$, $T_{zmax}$, $T_\infty$, and $T_d$ for each Paramotopy run and re-run is given in S1 Dataset, with further details in S1 Appendix.

**Certifying solutions.** The solutions reported by Bertini and Paramotopy are approximate. The software package alphaCertified, which implements Smale's $\alpha$-theory [77, 78, 94], can determine if an approximate non-singular solution $x^*$ to the polynomial system $f(x) = 0$ would converge under repeated application of Newton's method to an exact solution $\xi$. If so, we have a guarantee that the numerically obtained approximate solution is in the vicinity of an actual solution and we say that $x^*$ is a <u>certified approximate solution to $f(x) = 0$ with associated solution $\xi$</u>. This method cannot be applied to singular solutions, which do not behave well with respect to Newton's method. Accordingly, we sought to certify only the seven or five proper solutions identified for each parameter point at each value of $\sigma$.

To exploit this capability, we randomly chose 5% of each collection of proper solution sets associated with each Paramotopy run, yielding a total of $\sim 2.5 \times 10^7$ proper solution sets across 24 values of $\sigma$ (S1 Appendix), and used alphaCertified to certify each of these solutions. alphaCertified comes with a built-in module for sharpening non-singular solutions by additional Newton iterations, so we implemented a certify-and-sharpen procedure, in which any uncertified solution would undergo further sharpening before being passed to alphaCertified for another certification attempt.

With this procedure, we were able to certify almost every proper solution among the chosen $\sim 2.5 \times 10^7$ proper solution sets within five iterations of certification and four iterations of sharpening (totalling eight iterations of Newton's method). A tiny minority of 106 solution sets, all from parameter points sampled in $\mathcal{H}_5^*$, exhibited at least one uncertifiable proper solution, even after four iterations of sharpening. Among these 106 solution sets, 99 exhibited only one uncertifiable solution. Manual inspection of the alphaCertified output suggests that the uncertifiable solutions are in fact close to singular: applying successive iterations of Newton's method on these solutions, $(u^*, v^*)$, fails to show convergence in one or both of $|u^*|$ or $|v^*|$ (S1 Appendix). These apparently singular solutions evaded the tests imposed in Bertini and Paramotopy. However, they are extremely rare and were found only for the bistable region in $\mathcal{H}_5^*$. Accordingly, we do not believe they affect any of the quantitative conclusions we have drawn, even for $\mathcal{H}_5^*$.

In addition, we followed the same procedure to certify every proper solution, for every value of $\sigma$ in Eq 11, associated with each parameter point found to "blink" at some value of $\sigma$ (Results, Table 2). Each of these proper solutions was successfully certified within five iterations of certification and four iterations of sharpening.

A full description of our certification procedure, along with a discussion of uncertifiable solutions, is given in S1 Appendix.

## Confidence estimates for bistable volumes

Given an ILR sample $\mathcal{S} \subset \mathcal{H}$ of $N$ parameter points, it follows from Eq 13 that the unbiased volume estimator is given by

$$\widehat{V}_\sigma = \frac{1}{N} \sum_{\theta \in \mathcal{S}}^N \iota_\sigma(\theta) \, .$$

In other words, $\widehat{V}_\sigma$ is the sample mean of a sequence of independent and identically distributed Bernoulli random variables, $\iota_\sigma(\theta)$, with success probability $V_\sigma$. Therefore, the statistical properties of $\widehat{V}_\sigma$, in the limit of large $N$, are determined by the central limit theorem, as

$$\lim_{N \to \infty} \Pr\left(-\epsilon < \frac{\sqrt{N}(\widehat{V}_\sigma - V_\sigma)}{\delta} < \epsilon\right) = \Phi(\epsilon) - \Phi(-\epsilon) = 2\Phi(\epsilon) - 1$$

for any $\epsilon > 0$, where $\Phi$ is the standard normal cumulative distribution function, and $\delta = \sqrt{V_\sigma(1 - V_\sigma)}$ is the standard deviation of $\iota_\sigma$ over $\mathcal{H}$. Since the value of $\delta$ is unknown, we introduce the sample standard deviation of $\iota_\sigma$ over $\mathcal{S}$,

$$\widehat{\delta} = \sqrt{\frac{1}{N-1} \sum_{\theta \in \mathcal{S}} (\iota_\sigma(\theta) - \widehat{V}_\sigma)^2} \, ,$$

which converges to $\delta$ as $N \to \infty$. Therefore, we can write

$$\lim_{N \to \infty} \Pr\left(-\epsilon < \sqrt{N}(\widehat{V}_\sigma - V_\sigma)\widehat{\delta} < \epsilon\right) = 2\Phi(\epsilon) - 1$$

for any $\epsilon > 0$. In particular, for any $\alpha > 0$, we have

$$\lim_{N \to \infty} \Pr\left(|\widehat{V}_\sigma - V_\sigma| < \Phi^{-1}\left(1 - \frac{\alpha}{2}\right)\frac{\widehat{\delta}}{\sqrt{N}}\right) = 1 - \alpha.$$

Hence, the $100(1 - \alpha)\%$ confidence interval for $\widehat{V}_\sigma$ is given by

$$\widehat{V}_\sigma \pm \Phi^{-1}\left(1 - \frac{\alpha}{2}\right)\frac{\widehat{\delta}}{\sqrt{N}} \, ,$$

provided that $N$ is large.

## Confidence estimates for visibility ratios

In contrast to the volume estimator, $\widehat{V}_\sigma$, the estimator for the $K$-fold visibility ratio, $\widehat{\mathrm{vis}}(\mathcal{M}_\sigma, K)$, was computed by generating a random sample without replacement of $M = 20000$ pairs of bistable points,

$$(\theta^{(1)}, \mu^{(1)}), \quad \ldots, \quad (\theta^{(M)}, \mu^{(M)}) \, ,$$

from a finite population $\widehat{\mathcal{M}}_\sigma \times \widehat{\mathcal{M}}_\sigma$, where $\widehat{\mathcal{M}}_\sigma$ is a set of bistable points gathered with ILR sampling from $\mathcal{H}$, and evaluating

$$\widehat{\mathrm{vis}}(\mathcal{M}_\sigma, K) = \frac{1}{M} \sum_{j=1}^M v_K(\theta^{(j)}, \mu^{(j)}) \, .$$

Let $N = (\widehat{\mathcal{M}}_\sigma \times \widehat{\mathcal{M}}_\sigma)$. Provided that $M$ and $N - M$ are sufficiently large, an approximate 100

$(1 - \alpha)\%$ confidence interval for $\widehat{\mathrm{vis}}(\mathcal{M}_\sigma, K)$ is given by [95]

$$\widehat{\mathrm{vis}}(\mathcal{M}_\sigma, K) \pm \Phi^{-1}\left(1 - \frac{\alpha}{2}\right)\frac{\widehat{\delta}}{\sqrt{M}}\sqrt{1 - \frac{M}{N}},$$

where $\widehat{\delta}$ is the sample standard deviation of $v_K$ over $\mathcal{S}$,

$$\widehat{\delta} = \sqrt{\frac{1}{M-1}\sum\nolimits_{j=1}^{M}\left(v_K(\theta^{(j)}, \mu^{(j)}) - \widehat{\mathrm{vis}}(\mathcal{M}_\sigma, K)\right)^2},$$

and $\sqrt{1 - M/N}$ is a correction factor accounting for the finiteness of the population $\widehat{\mathcal{M}}_\sigma \times \widehat{\mathcal{M}}_\sigma$. It is important to note that $\widehat{\mathcal{M}}_\sigma \times \widehat{\mathcal{M}}_\sigma$ is fixed throughout this calculation. Running this calculation repeatedly with many distinct samples of size $M$ from $\widehat{\mathcal{M}}_\sigma \times \widehat{\mathcal{M}}_\sigma$, roughly $100(1 - \alpha)\%$ of the resulting confidence intervals would contain the value,

$$\frac{1}{N}\sum_{(\theta,\mu)\in\widehat{\mathcal{M}}_\sigma\times\widehat{\mathcal{M}}_\sigma} v_K(\theta, \mu).$$

As such, this confidence interval does not directly measure the accuracy of the estimate, $\widehat{\mathrm{vis}}(\mathcal{M}_\sigma, K)$, relative to the $K$-fold visibility ratio, $\mathrm{vis}(\mathcal{M}_\sigma, K)$; such a measurement would require, at a minimum, generating many distinct samples of bistable points, $\widehat{\mathcal{M}}_\sigma$, with which to compute $\widehat{\mathrm{vis}}(\mathcal{M}_\sigma, K)$.

We also note that $M \ll N$ for all values of $\sigma$, so that the correction factor becomes negligible. Thus, we simply compute the $100(1 - \alpha)\%$ confidence interval as

$$\widehat{\mathrm{vis}}(\mathcal{M}_\sigma, K) \pm \Phi^{-1}\left(1 - \frac{\alpha}{2}\right)\frac{\widehat{\delta}}{\sqrt{M}}.$$

## The VEGAS sampling algorithm

We outline here the VEGAS sampling algorithm described in the Results. Let $T$ be the total number of iterations, $N$ the size of each sample, $M$ the number of intervals, and $K$ the smoothing factor. We begin with an initial bistable sample, $\widehat{\mathcal{M}}_\sigma^{(0)} \subset [0.1, 10]^8$, gathered through some other sampling process (such as ILR sampling).

1. For each $j = 1, \ldots, 8$, partition the interval $[-1, 1]$ into $M$ bins $I_1^{(j)}, \ldots, I_M^{(j)}$ of equal length.

2. For each $t = 1, \ldots, T$, do the following:

   a. For each $j = 1, \ldots, 8$, do the following:

      i. Compute the histogram of values in the projection $\widehat{\mathcal{M}}_{\sigma,j}^{(t-1)} = \{\theta_j : \theta \in \widehat{\mathcal{M}}_\sigma^{(t-1)}\}$, according to the partition of $[-1, 1]$ given by $I_1^{(j)}, \ldots, I_M^{(j)}$.

      ii. Re-normalise the bin frequencies, $f_1^{(j)}, \ldots, f_M^{(j)}$, as follows:

      $$f_i^{(j)} \leftarrow \left\lceil\frac{Kf_i^{(j)}}{\#\widehat{\mathcal{M}}_\sigma^{(t-1)}}\right\rceil + 1,$$

      where $\lceil x \rceil$ denotes the ceiling of $x$, i.e., the least integer greater than $x$.

iii. Re-size the bins, $\{I_i^{(j)} = [a_i^{(j)}, b_i^{(j)}] : i = 1, \ldots, M\}$, to have length proportional to $f_i^{(j)}$.

iv. Sample a number, $r$, uniformly from $[0, 1]$, then partition the $N$ values to be sampled among the $M$ bins, so that the bin from which to generate the $j$th coordinate of the $n$th point, $\theta_j^{(n)}$, in $\mathcal{S}^{(t)}$ is given by

$$I(n, r) = I_i^{(j)} \,,$$

where $i$ is the least integer such that $i/M \geq (r + n)/N$.

v. For each $n = 1, \ldots, N$, do the following:

1. Sample a value, $s$, from the uniform distribution on $I(n, r)$.

2. Set $\theta_j^{(n)} = 10^s$.

b. Determine the bistable subset, $\mathcal{M}_\sigma \cap \mathcal{S}^{(t)}$, and update the total bistable sample as $\widehat{\mathcal{M}}_\sigma^{(t)} \leftarrow \widehat{\mathcal{M}}_\sigma^{(t-1)} \cup (\mathcal{M}_\sigma \cap \mathcal{S}^{(t)})$.

Step 2a, ii is a smoothing which we introduced for the following reason. If the bistable region, $\mathcal{M}_\sigma$, has very small volume relative to the bounding box then a sample $\widehat{\mathcal{M}}_\sigma$ can have many empty bins in the histograms over each projection. This can lead to heavy bias in future iterations of the sampling. With this in mind, we incorporated the smoothing factor, $K$, to re-normalise the bin frequencies and incremented each bin frequency by 1, so that each bin has a nonzero frequency. This slightly shifts the sampling probability along each parameter axis towards regions of lower sample density; this effect grows stronger as we accumulate more points in $\mathcal{M}_\sigma$. We fixed $M = 50$ and $K = 1000$ throughout our analysis, across all VEGAS samples generated for all values of $\sigma$. The values of $T$ and $N$ are described in the Results. A full description of our VEGAS implementation is given in S1 Appendix.

## Building the connectivity graph

We outline here the algorithm for building the connectivity graph, $\mathcal{G}_\Delta(\widehat{\mathcal{M}}_\sigma)$.

**Choosing $\Delta$.** We first consider the task of choosing $\Delta$ for a bistable sample $\widehat{\mathcal{M}}_\sigma$ coming from an ILR sample $\mathcal{S} \subset \mathcal{H}$. We reasoned that a suitable threshold for determining whether two points in $\widehat{\mathcal{M}}_\sigma$ are directly connected should be that, for any point $\theta \in \mathcal{S}$, the probability that there is a second point $\mu \in \mathcal{S}$ such that $d(\theta, \mu) \leq \Delta$ (where $d$ is Euclidean distance over logarithmic coordinates) is large, say, 0.99. That is, we want to choose $\Delta$ such that

$$\Pr\left(\exists \; \mu \in \mathcal{S} \; \text{such that} \; \mu \neq \theta \; \text{and} \; d(\theta, \mu) \leq \Delta\right) \approx 0.99 \,.$$

This is clearly 1 minus the probability that there exists no such point $\mu$. The probability that no point in $\mathcal{S}$ (other than $\theta$) is within distance $\Delta$ of $\theta$, assuming that $\mathcal{S}$ is an ILR sample, is given by

$$\left(1 - \frac{V_8(\Delta)}{V_\mathcal{H}}\right)^{\#\mathcal{S}-1} \,,$$

where $V_8(\Delta)$ is the volume of an 8-dimensional ball of radius $\Delta$,

$$V_8(\Delta) = \frac{\pi^4 \Delta^8}{24} \,.$$

So we want to choose $\Delta$ such that

$$1 - (1 - \frac{\pi^4 \Delta^8}{24 V_{\mathcal{H}}})^{\#\mathcal{S}-1} \approx 0.99 \, .$$

Rearranging, we find that a suitable value for $\Delta$ is given by

$$\Delta \approx (\frac{24 V_{\mathcal{H}}}{\pi^4}(1 - 0.01^{1/(\#\mathcal{S}-1)}))^{1/8} \, .$$

Now suppose that $\widehat{\mathcal{M}}_\sigma$ was built from an augmented VEGAS sample $\mathcal{S}_A$ obtained from an initial (ILR) sample $\mathcal{S}_I$. We can estimated an effective sample size, $N'$, for $\mathcal{S}_A$ as follows,

$$N' = \left( \frac{\text{bistable points in } \#\mathcal{S}_A}{\text{bistable points in } \#\mathcal{S}_I} \right) \mathcal{S}_I \, .$$

For instance, the VEGAS sample for $\sigma = 10$ was initialised with an initial set of 27508 bistable points from an ILR sample of $4 \times 10^6$ points (Fig 3), and, after $T = 6$ iterations, consisted of $6 \times 10^6$ points, of which 3208681 were bistable (Table 1). This gives an effective sample size of $N' \approx 4.67 \times 10^8$, which gives a value of $\Delta \approx 0.17$. Performing this calculation for each value of $\sigma$ given in Eq 15 and the corresponding bistable sample, we found that suitable values of $\Delta$ range between $\sim 0.10$ and $\sim 0.18$. Accordingly, we used $\Delta = 0.15$ for our connectivity analysis.

**Constructing $\mathcal{G}_\Delta(\widehat{\mathcal{M}}_\sigma)$.**   Given a set $\widehat{\mathcal{M}}_\sigma$ of bistable points and a constant $\Delta > 0$, chosen as above, the graph $\mathcal{G}_\Delta(\widehat{\mathcal{M}}_\sigma)$ is built as a spanning forest which contains each point in $\widehat{\mathcal{M}}_\sigma$ as a vertex, and connects two points with an edge if, and only if, the Euclidean distance between the two points is less than $\Delta$. Such a spanning forest is not unique.

Suppose $\widehat{\mathcal{M}}_\sigma = \{\theta^{(j)} : j = 1, \ldots, N\}$. The algorithm below returns an $N \times N$ adjacency matrix $A$ and a vector $L$ of $N$ component labels. The matrix satisfies $A_{ij} = 1$ if points $\theta^{(i)}$ and $\theta^{(j)}$ are connected by an edge and $A_{ij} = 0$ otherwise. The vector $L$ assigns each point to the label of its connected component.

1. Initialise *unvisited* $\leftarrow \widehat{\mathcal{M}}_\sigma$, *label* $\leftarrow 1$, $L_i \leftarrow 0$ for all $i = 1, \ldots, N$, and $A_{ij} \leftarrow 0$ for all $i, j = 1, \ldots, N$. Initialise an empty queue, *pointqueue*.

2. While *unvisited* is not empty, do the following:

   a. Choose a point $\theta^{(i)} \in$ *unvisited*.

   b. Remove $\theta^{(i)}$ from *unvisited*.

   c. Push $\theta^{(i)}$ onto the end of *pointqueue*.

   d. While *pointqueue* is not empty, do the following:

      i. Pop the first point $\theta^{(j)}$ from the start of *pointqueue*.

      ii. Set $L_j \leftarrow$ *label*.

      iii. For each $\theta^{(k)} \in$ *unvisited* such that $d(\theta^{(j)}, \theta^{(k)}) < \Delta$, do the following:

         1. Remove $\theta^{(k)}$ from *unvisited*.

         2. Set $A_{jk} \leftarrow 1$.

         3. Push $\theta^{(k)}$ onto the end of *pointqueue*.

   e. Update *label* $\leftarrow$ *label* + 1.

A complete description of our implementation of this algorithm is given in S1 Appendix.

**Refining the connectivity graph.** The refinement procedure was described in the main text. We used an approximation factor of $\epsilon = 0.001$ (Eq 16) throughout the analysis and sought to speed up this refinement process by increasing $K$ between iterations. Specifically, for each value of $\sigma$ listed in Eq 15, we updated $K$ as follows,

$$K \leftarrow \begin{cases} j + 1 & \text{if } \#C_2 > 10 \\ 3(j+1) & \text{if } 3 < C_2 \leq 10 \\ 10(j+1) & \text{otherwise,} \end{cases}$$

where $j = 0, 1, 2, \ldots$ is the iteration number, and $C_2$ is the second-largest component in the graph. This procedure yielded a single-component graph in two iterations for each value of $\sigma$ listed in Eq 15. Full details are given in S1 Appendix.

## Supporting information

**S1 Appendix. Supplemental methods.** This document provides a comprehensive description of our implementations of the methods described in the paper and guidelines for navigating the supplemental code and datasets. Supplemental figures: (A) Workflow for computing and parsing solutions with Paramotopy. Supplemental tables: (A) Seeds used to initialise the MATLAB pseudo-random number generator for sampling; (B) Seeds used to initialise the MATLAB pseudo-random number generator for VEGAS sampling; (C) Details of the refined connectivity graphs.
(PDF)

**S1 Code. Scripts used to process Paramotopy output.** A detailed description of each script in this collection is given in S1 Appendix.
(GZ)

**S1 Dataset. Summary of Paramotopy runs and associated Bertini settings.** This dataset contains: (1) a tab-delimited text file (`metadata.tsv`) enumerating all Paramotopy runs and re-runs performed in this analysis, along with the samples on which they were performed and the Bertini settings employed for each run and re-run; (2) a directory of tab-delimited text files (`thresholds/`) enumerating all Paramotopy runs and re-runs with their corresponding values of the thresholds $T_{zmin}$, $T_{zmax}$, $T_\infty$, and $T_d$; and (3) an XML file (`defaultprefs.xml`), passed as input into Paramotopy, that enumerates the default Bertini settings given in Table 3. See S1 Appendix for details.
(GZ)

**S1 File. Dataset DOIs.** All datasets are available on Mendeley Data under the given DOIs. See S1 Appendix for details.
(TSV)

## Author Contributions

**Conceptualization:** Jeremy Gunawardena.

**Formal analysis:** Kee-Myoung Nam, Benjamin M. Gyori.

**Methodology:** Kee-Myoung Nam, Benjamin M. Gyori, Silviana V. Amethyst, Daniel J. Bates, Jeremy Gunawardena.

**Software:** Kee-Myoung Nam, Benjamin M. Gyori, Silviana V. Amethyst, Daniel J. Bates.

**Supervision:** Jeremy Gunawardena.

**Writing – original draft:** Kee-Myoung Nam, Jeremy Gunawardena.

**Writing – review & editing:** Kee-Myoung Nam, Benjamin M. Gyori, Silviana V. Amethyst, Daniel J. Bates, Jeremy Gunawardena.

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
