## [Decision Letter · Decision Letter 0]

25 Feb 2020

Dear Dr. Gunawardena,

Thank you very much for submitting your manuscript "Robustness and parameter geography in post-translational modification systems" for consideration at PLOS Computational Biology. As with all papers reviewed by the journal, your manuscript was reviewed by members of the editorial board and by several independent reviewers. The reviewers appreciated the attention to an important topic. Based on the reviews, we are likely to accept this manuscript for publication, providing that you modify the manuscript according to the review recommendations.

Sincerely,

Pedro Mendes, PhD

Associate Editor

PLOS Computational Biology

Mark Alber

Deputy Editor

PLOS Computational Biology

[LINK]

Reviewer's Responses to Questions

**Comments to the Authors:**

Reviewer #1: In their article, Nam and coworkers, computationally explore the steady-state landscale of a dual phosphorylation cycle, and in particular scan for parameter areas of bistability (or better, areas that exhibit 3 steady states, as they don’t formally prove that these areas are bistable. They compare a system in which enzymes operate fully independent of their product, and systems in which the product inhibits the enzyme. More specifically, they derive an angebraic equation (a polynome) that describes the steady-state, and after scaling the parameters/total concentrations, they scan the parameter using sophisticated sampling algorithms to quantify areas of bistability. They find that bistability occurs only if enzyme (kinase) concentration is smaller then the substrate, and that product inhibition reduces the areas of bistability.

The major contribiton of the paper is two-fold: First, they provide a detailed algorithmic approach that can serve as a blueprint to explore mono/bistability in a complex reaction system systematically, and they provide detailed material in the supplement including scripts that may guide how to do this. Second, they find that bistability is less likely to occur when product inhibition is allowed.

Overall, the methods and analyses seem sound.

While I value these contributions, I still find it very difficult to really understand the implications. What are really the biologically significant findings? Can we understand it mechanistically, why bistability requires that substrate and enzyme concentrations are like described in the paper? I think the authors need to work on interpreting and also presenting their work such that it is more clear what the results mean in terms of signaling biology.

Reviewer #2: Review of

Robustness and parameter geography in post-translational modification systems

The authors build on their respective previous work on post-translational modification (PTM) systems by the Harvard lab and that of numerical algebraic geometry developers. They set out to create a general framework for (1) writing arbitrary enzyme kinetics of PTMs, in terms of polynomial functions Phi1 and Phi2, and (2) analysing the parameter space within Eq 10 using numerical algebraic geometry, and then quantifying the volume of the bistable region of this space. The authors find that the volume of bistability of the parameter geography varies significantly between different biological mechanisms.

Overall, the paper is well written, I found no typos while reading it. The additional information was useful for understanding the analysis performed.

I recommend it to be published, but first a few clarifications to improve the paper.

- The visibility ratio sounds like it is approximating geodesics. Can numerical algebraic geometry approximate this variety in parameter space, and provide some indication of the degree of this “visibility” curve?

- The ``grammar” sounds similar to studying the graph rather than the resulting equations, which is a feature of chemical reaction network theory. It would be useful to explain the connection when presenting chemical Eqs (1). For example, this grammar sounds similar to generalized mass action kinetics.

- The authors should discuss how their parameter geography compares to the work below, and whether the methods described below could provide insight to prove their conjectures:

Joshi and Shiu (SIAM Journal on Applied Mathematics), conditions on parameter

Conradi, Feliu, et al (PLOS Computational Biology) conditions on parameter

Harrington, Mehta et al (Communications in Computer and Information Science) stability and parameter geography

- For the parameter values that could not be certified, are these near the discriminant locus?

- It would be helpful to have more details in table captions.

Overall this is a good balance between application, theory and computation and recommend it for publication in PLOS Computational Biology.

Reviewer #3: Many biological systems depend upon parameters and it is important to understand the behavior of the system as the parameters change. The parameter space is partitioned into regions where the qualitative behavior of the steady-states of the system remain unchanged on each region. When the steady-state system is polynomial, tools from numerical algebraic geometry (such as Bertini, Paramotopy, and alphaCertified) can be used to analyze the "geography" of parameter space.

This paper considers a steady-state system which is polynomial in 2 variables dependent upon 8 parameters. By using random sampling of the set of parameters which are biologically meaningful, the authors analyze the "geography" of the parameter space. Overall, the setup for the computational experiments described in the paper appear to be reasonable while the scale of these computations is quite remarkable.

My only comment on the paper is in regards to decomposing the parameter space: Was there an attempt to symbolically compute the discriminant polynomial of (10)? Or compute it using numerical algebraic geometry? What is its degree?

Overall, this paper should be accepted with a minor revision based on the answers to these questions regarding the discriminant.

**Have all data underlying the figures and results presented in the manuscript been provided?**

Reviewer #1: Yes

Reviewer #2: Yes

Reviewer #3: Yes

PLOS authors have the option to publish the peer review history of their article (what does this mean?). If published, this will include your full peer review and any attached files.

Reviewer #1: No

Reviewer #2: No

Reviewer #3: No
---

## [Decision Letter · Decision Letter 1]

2 Apr 2020

Dear Dr. Gunawardena,

We are pleased to inform you that your manuscript 'Robustness and parameter geography in post-translational modification systems' has been provisionally accepted for publication in PLOS Computational Biology.

Best regards,

Pedro Mendes, PhD

Associate Editor

PLOS Computational Biology

Mark Alber

Deputy Editor

PLOS Computational Biology

Reviewer's Responses to Questions

**Comments to the Authors:**

Reviewer #3: The discriminant for the positive real roots is indeed algebraic -- the number of such roots can only change if a root becomes singular (classical discriminant) or by having a root intersect a coordinate axis. Lazard and Rouillier (JSC, 2007) call this the "minimal discriminant variety" for the corresponding problem -- see [Section 1.2, 67]. Nonetheless, I agree with the authors that this may be difficult to compute for this problem, so I am recommending acceptance.

**Have all data underlying the figures and results presented in the manuscript been provided?**

Reviewer #3: Yes

PLOS authors have the option to publish the peer review history of their article (what does this mean?). If published, this will include your full peer review and any attached files.

Reviewer #3: No

---

## [Editor Report · Acceptance letter]

23 Apr 2020

PCOMPBIOL-D-19-02084R1 

Robustness and parameter geography in post-translational modification systems

Dear Dr Gunawardena,

I am pleased to inform you that your manuscript has been formally accepted for publication in PLOS Computational Biology. Your manuscript is now with our production department and you will be notified of the publication date in due course.

With kind regards,

Laura Mallard
